# From Models to Systems: A Comprehensive Survey of Efficient Multimodal Learning

Pan Wang[1][*][†], Siwei Song[8*], Hui Ji[1*], Siqi Cao[3*], Heng Yu[9], Zhijian Liu[6], Huanrui Yang[7], Yingyan (Celine) Lin[5], Beidi Chen[3], Mohit Bansal[2], Xiaoming Liu[2], Pengfei Zhou[1], Ming-Hsuan Yang[4], Tianlong Chen[2], Jingtong Hu[1]

[1]University of Pittsburgh    [2]UNC Chapel Hill    [3]Carnegie Mellon University    [4]UC Merced    [5]Georgia Tech
[6]UCSD    [7]University of Arizona    [8]New York University    [9]Stanford University

**Reviewed on OpenReview:** https://openreview.net/forum?id=yfTU8FTS2Z

## Abstract

The rapid expansion of multimodal models has surfaced formidable bottlenecks in computation, memory, and deployment, catalyzing the rise of Efficient Multimodal Learning (EML) as a pivotal research frontier. Despite intensive progress, a cohesive understanding of *what*, *how*, and *where* efficiency is manifested across the learning stack remains fragmented. This survey systematizes the EML landscape by introducing the first structured, model-to-system taxonomy. We distill insights from over 300 seminal works into three hierarchical levels—*model*, *algorithm*, and *system*—addressing architectural parsimony, execution refinement, and hardware-aware orchestration, respectively. Moving beyond a purely categorical review, we offer a methodological synthesis of the vertical synergies between these layers, elucidating how cross-layer co-design contributes to the fundamental "Efficiency-Utility-Privacy" trade-off. Through an integrative case study of Multimodal Large Language Models (MLLMs), we trace the field's evolutionary trajectory from initial structural adjustments to modern full-stack resource orchestration. Furthermore, we provide a holistic discussion and application-specific optimization blueprints for diverse domains and posit a paradigm shift toward self-regulating intelligence, where efficiency is an intrinsic, emergent property of the model's fundamental design rather than a post-hoc constraint. Finally, we present open challenges and future directions that will define the trajectory of EML research. This survey establishes a structured framework for multimodal systems that are not only high-performing and generalizable but natively efficient and ready for ubiquitous deployment. A continuously updated version is available at https://github.com/pwang322/Efficient-Multimodal-Learning-Survey.

## 1 Introduction

The paradigm shift toward multimodal learning has revolutionized artificial intelligence, enabling systems to jointly perceive, align, and reason over heterogeneous signals such as vision, language, audio, and sensor data (Baltrušaitis et al., 2018; Mo et al., 2024; 2023). This unification underpins critical advances in domains ranging from embodied robotics and autonomous driving to precision healthcare (Jin et al., 2025). However, this scaling success faces a formidable bottleneck: computational inefficiency. The quadratic complexity and massive parameter counts of modern multimodal transformers demand exorbitant memory and energy resources, often precluding deployment in real-time or resource-constrained environments. As the field moves to democratize these models beyond high-end clusters, establishing a systematic understanding of Efficient Multimodal Learning (EML) has become a critical academic and industrial frontier.

---

[*]Equal Contribution.
[†]Corresponding author: Pan Wang (pan.wang@pitt.edu).

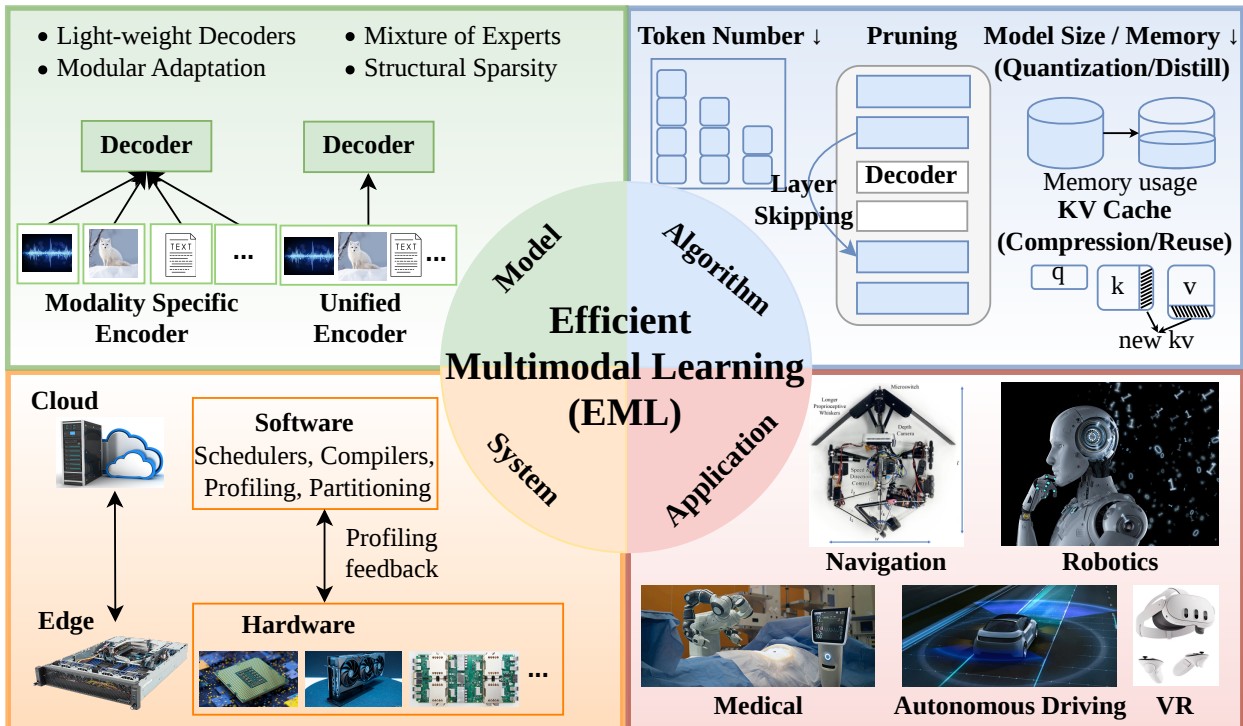

Figure 1: Overall landscape of Efficient Multimodal Learning (EML), organized across three interconnected levels—Model, Algorithm (Compression & Acceleration), and System—that jointly optimize architectural design, computation, and deployment, with representative applications illustrated. All these methods will facilitate downstream applications such as Autonomous driving, VR, Robotics, and so on.

## 1.1 The Multimodal Efficiency Challenge

Unlike unimodal efficiency, which primarily optimizes homogeneous data streams, multimodal efficiency faces the fundamentally harder task of orchestrating heterogeneous signals with disparate semantics, resolutions, and temporal dynamics. This orchestration must navigate a highly diverse operational scope: while EML strictly requires multimodal inputs, it accommodates both unimodal output predictions for understanding tasks and continuous multimodal output streams for generation, across data conditions ranging from strictly paired to fully disjoint sets. Consequently, the core computational bottleneck extends far beyond general model scaling; it is intrinsically rooted in the specific efficiency prerequisites of these paradigms. For multimodal understanding, optimization hinges on low-complexity encoders and cross-modal alignment, whereas generation introduces the profound overhead of autoregressive decoding and detokenization. Crucially, EML seeks to bridge discrete symbolic reasoning with high-density, continuous perceptual data. Grounding abstract logic in the tangible world via resource-intensive sensory streams dictates that optimization can no longer be isolated to a single layer. Instead, it mandates a holistic, full-stack orchestration—spanning architectural design, algorithmic compression, and hardware-aware system execution. Ultimately, this evolution marks a profound paradigm shift: moving from purely accuracy-centric scaling toward intelligence that is natively efficient, resource-adaptive, and sustainably deployable at scale.

## 1.2 The MAS Taxonomy

Despite the explosion of research, the path to multimodal efficiency remains obscured by fragmented techniques. While recent surveys provide excellent coverage of MLLM capabilities (Zhang et al., 2024a), specific compression mechanisms (Yao et al., 2025), or isolated optimization stages (Jin et al., 2025; Shinde et al., 2025), they typically adopt a layer-specific approach. This fragmented perspective fails to answer a fundamental structural question: *Where exactly in the multimodal stack can efficiency be injected, and how do*

*these injections interact?* To bridge the critical gap between high-level architectural designs and underlying physical execution constraints, we propose the *Model-Algorithm-System* (MAS) taxonomy, as illustrated in Figures 1 and 2. To the best of our knowledge, this is the first survey to introduce a full-stack taxonomy for EML. By synthesizing over 300 studies, this MAS framework maps optimizations to their precise locus within the computing stack.

- **Model-level:** reshaping architectural topology to define *what* to compute—encompassing modality-specific or unified encoders, sparse expert routing, structural decoding, and modular adaptation.

- **Algorithm-level:** modulating information flow to determine *how* to compute—via token compression, pruning, quantization, distillation, speculative decoding, and cache reuse.

- **System-level:** orchestrating physical execution to decide *where* and *when* to compute—integrating cache management, edge–cloud collaboration, latency-aware scheduling, and hardware-software co-design.

## 1.3 Survey Methodology

To ensure a systematic, comprehensive, and reproducible review of EML, we adopt the following literature synthesis methodology:

- **Search Strategy and Temporal Coverage:** We queried major databases (e.g., arXiv, Google Scholar, DBLP) for literature published between 2020 and 2026, capturing the evolution from early cross-modal alignment to modern Large Multimodal Models (MLLMs). We prioritized top-tier AI venues (e.g., ICLR, ICML, NeurIPS, CVPR, ICCV, ECCV, ACL, EMNLP, NAACL, ACM MM, and AAAI), as well as recent works on arXiv, using keywords centered on multimodal efficiency (e.g., "Compressed Multimodal", "Efficient MLLM", "Multimodal Pruning/Quantization/Distillation", and "Hardware-aware Systems").

- **Scope and Selection Criteria:** Included works must demonstrate quantifiable resource reductions (e.g., FLOPs, inference latency, memory footprint, or energy). We explicitly excluded research focused solely on general representation learning, pure accuracy scaling, or unimodal optimization, unless these methods seamlessly integrate efficiency-driven mechanisms (e.g., sparsity-inducing alignment) or serve as foundational modality-specific encoders within a broader multimodal pipeline.

- **Dimensions over Numerical Leaderboards:** A core objective of this survey is to map the structural design space of EML rather than compile an empirical numerical leaderboard. Because raw performance metrics (e.g., latency or memory reduction) are highly entangled with heterogeneous hardware backends, task formulations, and deployment constraints, direct numerical comparisons are often scientifically incomparable and misleading. Consequently, our synthesis prioritizes methodological dimensions and operational alignments.

## 1.4 Contributions

By analyzing the vertical synergies across the Model, Algorithm, and System layers, we demonstrate how the convergence of perception, execution, and scheduling mitigates the fundamental Efficiency-Utility-Privacy trilemma. We establish application-specific optimization blueprints for diverse domains—from affective computing to spatial understanding and reasoning—and posit a paradigm shift toward self-regulating intelligence. In this nascent regime, efficiency is reframed not as a post-hoc constraint, but as an intrinsic, emergent property of the model's fundamental design. By delineating these critical open challenges and future directions, we offer the community a coherent guide to realizing natively efficient and scalable multimodal intelligence.

This survey makes the following contributions:

- **Unified Taxonomy:** We present the blueprint integrating model, algorithm, and system efficiency into a holistic MAS framework for EML.

- **Cross-Level Analysis:** We deconstruct the dependencies between architectural sparsity, algorithmic compression, and system orchestration, offering a comprehensive view of resource-aware intelligence.

- **MLLMs Synthesis:** We synthesize recent breakthroughs and evolution in efficient MLLMs as a critical convergence within the MAS framework, where vertical integration empowers scalable, real-world multimodal intelligence.

- **Future Roadmap:** We identify emerging applications and open questions, showing future directions toward sustainable, adaptive, and deployable multimodal learning.

The remainder of this survey is structured as follows. Sections 2–4 systematically examine efficiency across the model, algorithm, and system levels. Building upon this foundation, Section 5 analyzes the evolution of efficient MLLMs, followed by discussions on real-world applications (Sec. 6) and holistic structural insights (Sec. 7). Finally, Section 8 outlines open research challenges. Additionally, Appendix 9 provides extensive supplementary materials, including a detailed comparison with recent surveys to explicitly position our contributions (Table 5, Appendix C), a comprehensive catalog of recent multimodal models (Appendix B), and a practical case study for edge EML systems (Appendix A).

## 2 Model

As shown in Fig. 3, model-level efficiency fundamentally reshapes the architectural topology to define *what* to compute. Efficient architectures seek to minimize redundant processing while preserving alignment, interaction, and representational richness. They reshape computation through explicit structural choices—encoder specialization, unified encoders, sparsity-aware routing, structural decoding, and modular adaptation—each offering a pathway to achieve scalable and expressive multimodal learning under limited budgets.

### 2.1 Modality-specific Encoders

Modality-specific encoders serve as the foundational feature extractors in multimodal systems, typically inheriting from mature unimodal architectures. Rather than cross-modal interactions, this subsection specifically reviews the internal efficiency optimizations within these individual modality streams (e.g., vision, text, or audio encoders) to reduce the foundational computational overhead.

**Image.** The evolution of vision encoders reflects a continuous search for efficient topologies that balance perceptual fidelity with computational constraints. Early efficiency-oriented designs, such as MobileNet (Howard, 2017), ShuffleNet (Zhang et al., 2018), and EfficientNet (Tan & Le, 2019), are CNNs that were first optimized for image recognition efficiency and representation quality. They mitigate computational redundancy via depthwise separable convolutions or bottleneck residual blocks that optimize the ratio between spatial resolution and channel depth. While effective for local feature extraction, their lack of global context prompts the development of hybrid architectures like MobileViT (Mehta & Rastegari, 2021) and CoAtNet (Dai et al., 2021b), which synergize convolutional efficiency with Transformer expressivity. Pushing this structural evolution further, PoolFormer (Yu et al., 2022b) demonstrates that simple pooling operations can replace complex attention mechanisms within a "MetaFormer" architecture, achieving efficiency through pure topological simplification.

The subsequent shift to Vision Transformers (ViT) (Dosovitskiy et al., 2020) fully enables global representation but incurs quadratic complexity. To address this, hierarchical variants like Swin Transformer (Liu et al., 2021) reintroduce shifted-window attention to linearize complexity while preserving local priors. Simultaneously, Masked Autoencoders (MAE) (He et al., 2022) and BEiT v2 (Peng et al., 2022) resolve the scalability bottleneck by turning masked image modeling into an efficiency primitive, enabling large-scale pretraining with reduced overhead. More recently, alternatives have emerged to challenge the dominance of attention entirely: Mamba-based backbones (Gu & Dao, 2024; Zhu et al., 2024a) use selective state-space models to capture long-range dependencies with linear sequence complexity O(N), while Kolmogorov–Arnold Networks (KANs) (Liu et al., 2024b) replace fixed nonlinearities with learnable spline-based functions and have been reported to achieve favorable parameter-efficiency and neural scaling behavior relative to MLP-style blocks.

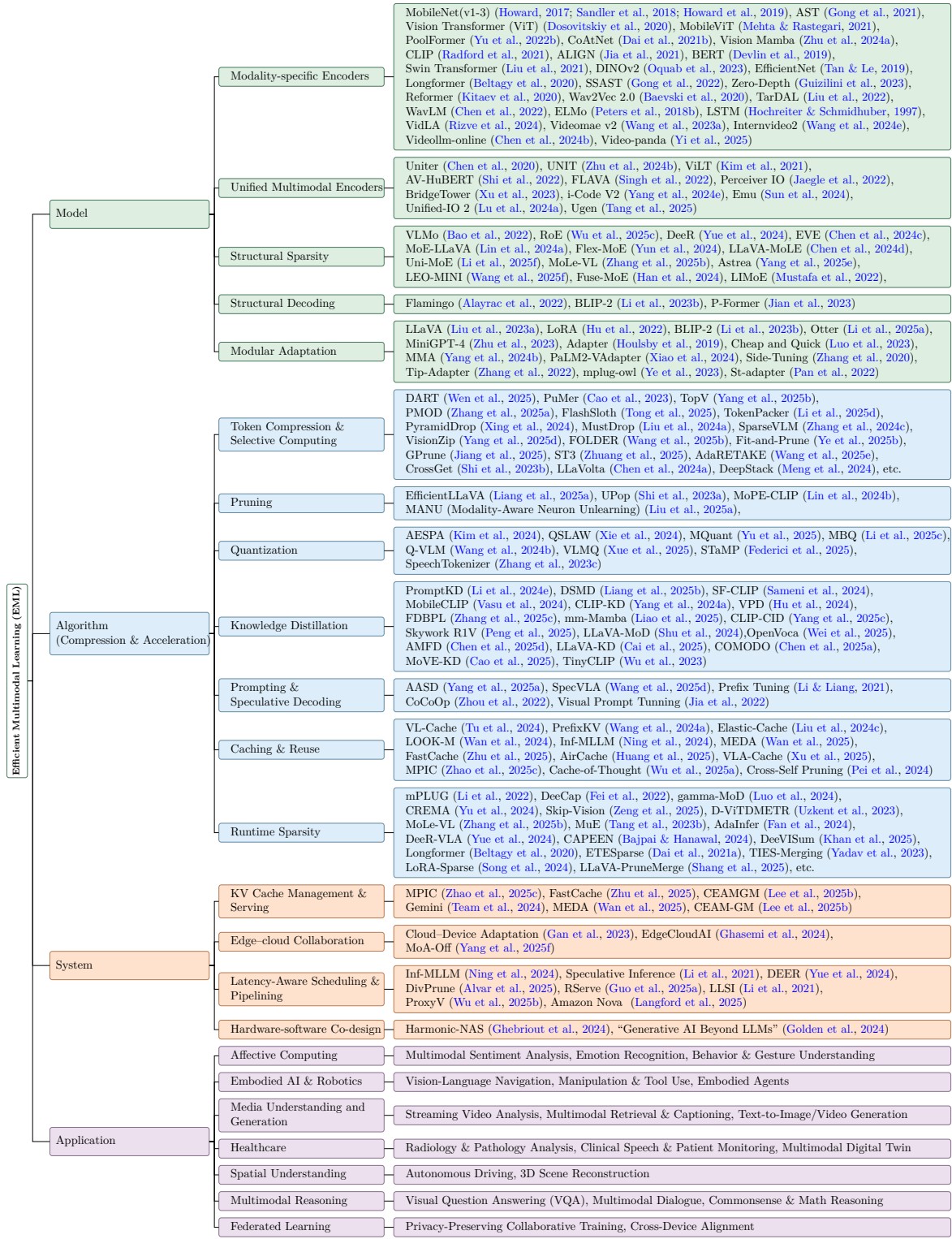

Figure 2: The MAS taxonomy for EML with representative works.

Ultimately, the evolution of visual representations has shifted from purely structural optimization to cross-modal semantic alignment. This paradigm is driven by dual-stream multimodal models like CLIP (Radford et al., 2021) and ALIGN (Jia et al., 2021), which project visual and textual modalities into a unified semantic

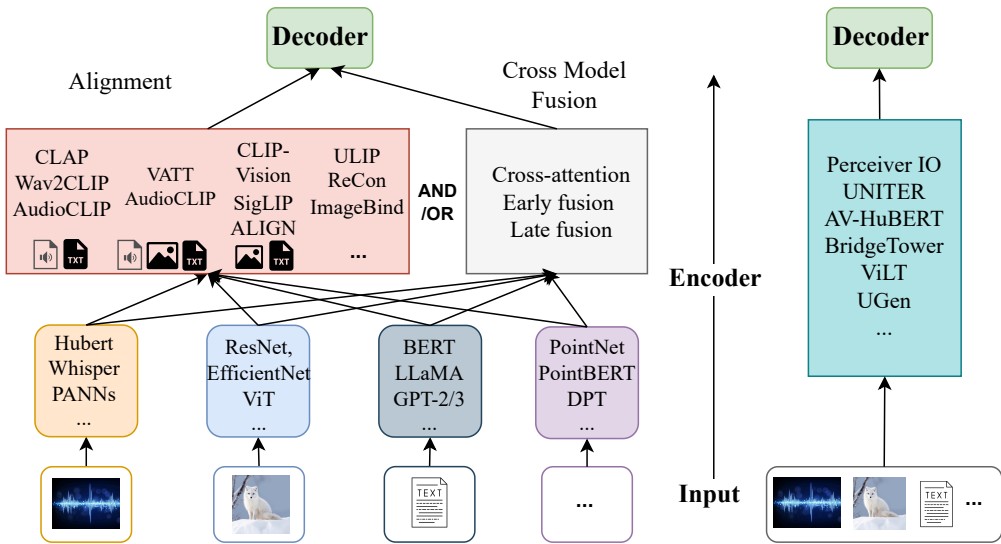

Figure 3: Structural paradigms of multimodal encoders. The taxonomy contrasts (left) decoupled modality-specific pipelines utilizing post-hoc alignment or fusion mechanisms with (right) natively unified encoders that collapse heterogeneous signals into a shared parameterized core. This architectural evolution reflects a shift toward functional consolidation, where unification acts as a structural prerequisite for efficiency.

space via the InfoNCE objective:

$$\mathcal{L}_{\text{InfoNCE}} = -\frac{1}{N} \sum_{i=1}^{N} \log \frac{\exp\big(\text{sim}(v_i, t_i)/\tau\big)}{\sum_{j=1}^{N} \exp\big(\text{sim}(v_i, t_j)/\tau\big)}, \tag{1}$$

where $\text{sim}(\cdot)$ denotes cosine similarity, $\tau$ is a temperature parameter, and $N$ is the number of samples. Building on this foundation, SigLIP (Zhai et al., 2023) identifies the softmax normalization as a scalability bottleneck and replaces it with a pairwise sigmoid loss, decoupling memory usage from batch size. Complementing these global alignment methods, dense self-supervised vision-only encoders like the DINO series (Caron et al., 2021; Oquab et al., 2023; Siméoni et al., 2025) provide fine-grained visual features essential for multimodal understanding. This paradigm transforms visual encoders from static classifiers into flexible foundations for open-vocabulary multimodal systems.

**Video.** To address the temporal redundancy and computational expansion of multi-frame inputs, video-specific architectures prioritize structural sparsity and hierarchical management over traditional 3D backbones. VideoMAE V2 (Wang et al., 2023a) establishes a scalable paradigm using extreme masking (90%), proving that billion-parameter models can be trained efficiently by reconstructing only a fraction of video tokens. Similarly, InternVideo2 (Wang et al., 2024e) employs a progressive training approach to scale encoders to 6B parameters, unifying masked modeling with cross-modal contrastive learning to enhance long-context comprehension.

Efficiency is further achieved by optimizing temporal dependency extraction without heavyweight backbones. VidLA (Rizve et al., 2024) utilizes hierarchical temporal tokens to capture multi-scale motion within a simple two-tower structure initialized from 2D foundations. More radically, Video-Panda (Yi et al., 2025) introduces an encoder-free Spatio-Temporal Alignment Block (STAB), reducing visual parameters by 6.5× while maintaining competitive Video-QA performance. Unlike structural sparsity or modular adaptation which refine existing backbones, Video-Panda is categorized as a modality-specific encoder because its STAB replaces the traditional pre-trained foundation model entirely, acting as a lightweight, primary visual-processing core tailored for spatio-temporal alignment. Moving beyond offline clips, the LIVE framework in VideoLLM-online (Chen et al., 2024b) optimizes the inference pipeline for continuous streaming. By converting temporal annotations into a streaming dialogue format, it enables real-time interaction (10+ FPS), allowing multimodal systems to sustain long-context reasoning over continuous video feeds.

**Text.** The evolution of text encoding traces a cyclical trajectory: moving from memory-efficient recurrence to parallelized attention, and finally converging on architectures that unify the strengths of both to support massive multimodal contexts. The foundational era relies on Recurrent Neural Networks (RNNs), where architectures like LSTM (Hochreiter & Schmidhuber, 1997) and GRU (Cho et al., 2014) maintain hidden states to capture temporal dependencies. While deep contextualized representations like ELMo (Peters et al., 2018a) demonstrate the representational power of this paradigm, the inherent sequentiality of recurrence prohibits parallel training, creating a fundamental barrier to distributed training on modern hardware. Efficiency-oriented variants, such as IndRNN (Li et al., 2018) and LightRNN (Li et al., 2016), attempt to mitigate this by decoupling matrix operations or compressing vocabularies, yet the underlying throughput bottleneck persists.

The introduction of the Transformer (Vaswani et al., 2017) and BERT (Devlin et al., 2019) breaks this serial constraint by enabling fully parallel context aggregation. However, this architectural shift exchanges sequential latency for quadratic computational complexity ($O(N^2)$), which becomes prohibitive when processing long sequences of interleaved text tokens. To reconcile global context with computational viability, structural variants dismantle the dense attention matrix: Longformer (Beltagy et al., 2020) and BigBird (Zaheer et al., 2020) introduce sparse window mechanisms to reduce complexity to linear time, while Reformer (Kitaev et al., 2020) employs locality-sensitive hashing (LSH) to approximate global interactions. Simultaneously, approaches like Linformer (Wang et al., 2020) demonstrate attention matrices are low-rank, allowing for projection-based approximations that further compress the computational footprint.

As decoder-only LLMs, including LLaMA (Touvron et al., 2023), Mistral (Jiang et al., 2023), and Qwen (Bai et al., 2023), become the foundation for modern VLMs like LLaVA (Liu et al., 2023a), MiniGPT-4 (Zhu et al., 2023), and Qwen-VL (Team, 2023), the frontier shifts toward revisiting recurrence to handle the explosive growth of multimodal context windows. Recent architectures, including Linear Transformers (Katharopoulos et al., 2020) and state-space models (SSMs) like TextMamba (Zhao et al., 2025b), abandon the standard softmax attention mechanism entirely. By combining the parallelizable training of Transformers with the constant inference memory of RNNs ($O(1)$), these designs unlock a critical capability for multimodal learning: the ability to sustain effectively infinite context windows. This transforms text encoders from a computational bottleneck into a scalable semantic anchor, capable of maintaining long-term dialogue history and reasoning over extensive descriptive inputs essential for multimodal systems.

**Audio.** The evolution of audio encoders parallels that of vision, transitioning from rigid convolutional priors to unified, efficiency-aware tokenization that aligns seamlessly with broader multimodal architectures. Early efficiency-oriented designs rely on architectural inductive biases: by treating log-Mel spectrograms as 2D image-like signals, convolutional encoders such as VGGish (Hershey et al., 2017) and CNN14 (Kong et al., 2020) leverage local connectivity to extract harmonic patterns. While effective, this approach defines efficiency primarily through parameter sharing and locality, often overlooking the temporal continuity intrinsic to acoustic signals. To overcome this and leverage massive unlabeled data, the paradigm shifts toward self-supervised learning (SSL) within hybrid topologies. Foundational frameworks like Wav2Vec 2.0 (Baevski et al., 2020) and HuBERT (Hsu et al., 2021) combine lightweight convolutional front-ends for local feature extraction with Transformer backbones for global context. By solving contrastive or masked unit prediction tasks, these methods redefine efficiency through the lens of data scalability, producing robust representations that generalize across tasks with minimal supervision.

A critical turning point toward architectural unification arrives with the adoption of patch-based masked modeling, inspired by Vision Transformers. The Audio Spectrogram Transformer (AST) (Gong et al., 2021) and SSAST (Gong et al., 2022) eliminate the convolutional hierarchy, treating spectrogram patches as discrete tokens for global self-attention. This shift not only simplifies the design space but also aligns audio encoding structurally with visual and textual modalities, facilitating cross-modal transfer and unified pretraining. However, the quadratic complexity of global attention poses a bottleneck for long-form audio processing. Addressing this, recent architectures like Audio Mamba (Yadav & Tan, 2024) adopt selective SSMs to bypass the attention mechanism. By capturing long-range dependencies with strict linear complexity ($\mathcal{O}(N)$) and constant inference memory, these models naturally align with the continuous nature of sound.

Beyond architectural shifts, the integration of speech into Large Language Models (LLMs) has catalyzed the development of both continuous and discrete tokenizers. While SSL models provide robust standalone representations, weakly-supervised continuous encoders like Whisper (Radford et al., 2023) yield highly semantic features and are now extensively utilized as plug-and-play speech encoders for text-audio and video-audio LLMs. Concurrently, to navigate the high bitrate of raw waveform data efficiently, the field has increasingly embraced discrete tokenization. Frameworks such as SpeechTokenizer (Zhang et al., 2023c) and Mimi (Défossez et al., 2024) (introduced with Moshi) compress acoustic streams into hierarchical discrete codes, disentangling semantic content from acoustic details. This trajectory culminates in a vital capability for efficient multimodal learning: translating high-fidelity acoustic streams into compact, manageable token vocabularies, thereby establishing a scalable foundation for real-time, omni-sensory understanding.

**Beyond Canonical Modalities.** Beyond vision, text, and audio, modalities such as thermal imagery, depth sensing, and time-series data extend multimodal learning into domains characterized by low resolution, sparsity, or temporal irregularity. Although heterogeneous, the evolution of their encoders reveals a convergent trajectory: moving from rigid priors to flexible, computation-aware abstractions.

**Thermal** imagery captures radiometric intensity rather than clear texture, often resulting in low-contrast, high-noise data. To process this efficiently, architectures must isolate informative features from clutter. Early designs such as TarDAL (Liu et al., 2022) employ dual-stream topologies to disentangle target semantics from noise via sub-networks. More recently, approaches like FW-SAT (Jiang & Chen, 2024) transition to ViT-based backbones but restrict computation through local window attention. This design mitigates the quadratic cost of global modeling while focusing resources on informative regions, preserving the structural details vital for interpreting low-quality thermal inputs without incurring the overhead of full self-attention.

**Depth** sensing provides explicit geometric cues but frequently suffers from sparse or missing measurements due to sensor limitations. To balance structural fidelity with computational cost, the field converges on hybrid architectures. Frameworks like Lite-Mono (Zhang et al., 2023a) and MonoDETR (Zhang et al., 2023b) integrate lightweight convolutions for high-frequency surface completion and Transformer blocks for global geometric reasoning. This hybrid topology effectively leverages the efficiency of convolutions for handling local sparsity (filling "holes" in the depth map) while reserving expensive attention operations solely for establishing long-range scale consistency.

**Time-series** data pose unique challenges regarding irregular sampling and extremely long-range dependencies. While RNNs capture local trends, their sequential nature prevents parallel training on massive datasets. To address this, efficient Transformers like Informer (Zhou et al., 2021) introduce sparse attention mechanisms, such as ProbSparse attention, to approximate global context with sub-quadratic cost ($O(N \log N)$). Most recently, the focus shifts to continuous-time architectures: State-space models like Mamba (Gu & Dao, 2024) and Liquid-S4 (Hasani et al., 2023) model temporal evolution via linear recurrence. By processing massive horizons with strict linear complexity ($O(N)$), these models resolve the memory bottleneck of Transformers, establishing a scalable paradigm for long-term forecasting and sequential reasoning.

## 2.2 Unified Multimodal Encoders

Unified multimodal encoders aim to collapse redundant, modality-specific pipelines into a shared computational backbone (Wang et al., 2022). Instead of maintaining parallel branches for each signal, these architectures introduce a centralized parameterized core—such as a joint Transformer trunk—that processes heterogeneous tokens within a single vector space. This paradigm shifts the definition of efficiency from simple resource reduction to functional consolidation, where cross-modal interactions occur deeply and repeatedly, turning unification itself into a mechanism for parameter and inference efficiency.

**From Dual-stream Fusion to Shared Trunks.** Early efforts focus on integrating visual and textual streams within a single Transformer, which is introduced for language processing in NLP (Vaswani et al., 2017; Li et al., 2019). Models like UNITER (Chen et al., 2020), ViLT (Kim et al., 2021), and BridgeTower (Xu et al., 2023) discard heavy modality-specific extractors, instead use ViT-style encoding image patches and text tokens directly through a shared attention backbone. Vlmo (Bao et al., 2022) refines this by sharing the self-attention trunk while using modality-specific experts to handle distinct feature distributions. These

works establish the principle of parameter sharing as efficiency, proving that cross-modal understanding does not require independent feature hierarchies.

**Multi-sensory Unification.** Subsequent frameworks extend this unified paradigm to include audio and video. AV-HuBERT (Shi et al., 2022) generalizes SSL by jointly masking and predicting clustered audio–visual units, achieving strong recognition accuracy with orders of magnitude less labeled data. FLAVA (Singh et al., 2022) further scales this by employing a single Transformer to process image, text, and audio tokens, demonstrating that cross-modal co-training acts as an implicit regularizer. Expanding this to any-to-any encoding, OmniVec (Srivastava & Sharma, 2024) consolidates depth, video, audio, text and other modalities into a single universal trunk, while AudioPaLM (Rubenstein et al., 2023) achieves extreme efficiency by treating speech tokens as a specialized linguistic variant within a shared text-based vocabulary. This holistic approach reduces the total pretraining compute compared to maintaining separate unimodal models, validating unification as a path to scalability.

**Latent-Core and Autoregressive Unification.** Recent designs unify capacity through latent bottlenecks or sequence-level modeling. Perceiver IO (Jaegle et al., 2022) and Perceiver-VL (Tang et al., 2023c) encode arbitrary modalities via a fixed-size latent array, decoupling computational cost from input size and resolution. Alternatively, UNIT (Zhu et al., 2024b) maintains lightweight modality-specific heads during pretraining but merges them at inference for a single shared encoder. More radically, autoregressive models such as Emu (Sun et al., 2024), Unified-IO 2 (Lu et al., 2024a), i-CodeV2 (Yang et al., 2024f), UGen (Tang et al., 2025), and Grok-1.5V (xAI / Grok team, 2024) embrace generative unification. To achieve generative unification, these models map diverse signals into a shared discrete space. Images are typically decomposed into spatial patches and quantized via VQ-VAE (Van Den Oord et al., 2017) or VQGAN codebooks (Esser et al., 2021). Speech and Audio are processed by transforming raw waveforms into latent representations—either through spectral features or neural audio codecs like EnCodec (Défossez et al., 2022)—which are then discretized into temporal tokens. For video, 3D tubelet (Arnab et al., 2021) embedding layers capture spatiotemporal dependencies before quantization. This transforms continuous signals into a sequence of discrete indices that are interleaved with text tokens, allowing the language model to optimize a unified next-token prediction objective:

$$\mathcal{L}_{\mathrm{AR}} = -\sum_{t=1}^{T} \log p(y_t \mid y_{<t}, \mathrm{ctx}), \tag{2}$$

where $y_t$ denotes the target token at step $t$, $y_{<t}$ represents the multimodal history, and ctx is the context. This formulation enables seamless conditioning and generation across modalities, establishing sequence modeling as the universal interface for multimodal efficiency.

**Structural Trade-offs in Encoder Design.** The choice between modality-specific and unified architectures represents a fundamental trade-off in computational allocation. Modality-specific encoders prioritize front-end specialization. By leveraging strong inductive biases and specialized pre-training, they capture high-fidelity features before multimodal fusion occurs. This decoupled approach is computationally efficient for storage and allows for the easy integration of state-of-the-art unimodal backbones. In contrast, Unified Encoders prioritize parameter sharing and early interaction. By processing all modalities through a common transformer stack, these models reduce infrastructure duplication and enable deeper cross-modal reasoning within the same latent space. Modality-Specific Encoders are favored in domain-specialized applications where the input data has a high information density or unique statistical properties that general-purpose transformers might overlook. While Unified Encoders are preferred for general-purpose reasoning and any-to-any generation tasks.

## 2.3 Structural Sparsity

Structural sparsity enforces efficiency by utilizing conditional computation to activate only a subset of parameters during training or inference. Unlike pruning (which permanently removes weights), structural sparsity embeds dynamic routing directly into the model architecture, allowing systems to decouple total capacity from active computation. While this paradigm originates from border sparse-conditional compu-

tation and MoE (Shazeer et al., 2017) research, recent multimodal models increasingly adapt it to reduce cross-modal interference and support unified conditional computation. In multimodal contexts, this enables models to scale to billions of parameters while maintaining the inference footprint of much smaller networks, dynamically allocating resources based on input complexity.

**Modality-Specialized Expert Routing.** Early applications of Mixture-of-Experts (MoE) in multimodal learning focus on mitigating cross-modal interference. Model architectures like VLMo (Bao et al., 2022) and LIMoE (Mustafa et al., 2022) introduce modality-aware routing, coupling shared attention mechanisms with modality-specific experts. By directing image and text tokens to distinct feed-forward networks (FFNs), these models achieve disentangled representation learning, proving that sparsity can enhance expert specialization while reducing the computational redundancy of monolithic transformers.

**Unified Semantic Routing.** Subsequent frameworks, such as Uni-MoE (Li et al., 2025f), have advanced from rigid modality partitioning to unified, content-driven routing. Here, experts are shared across modalities and activated dynamically based on token-level semantic complexity. This shift allows the architecture to adaptively allocate capacity—using more experts for complex reasoning tokens and fewer for simple patches—transforming sparsity from a static routing rule into a responsive, content-aware mechanism.

**Budget-Aware Elasticity.** Recent frameworks such as LEO-MINI (Wang et al., 2025f) and Flex-MoE (Yun et al., 2024) extend sparsity to resource-constrained deployment. Rather than maximizing capacity, they prioritize compute elasticity, employing hierarchical routing or mixed-rank experts to satisfy strict memory or latency budgets. Models like NVILA (Liu et al., 2025b) and SmolVLM (Marafioti et al., 2025) further optimize this by pruning token pathways, effectively creating any-budget architectures that dynamically adjust their active parameter set to fit the available hardware envelope.

From a computational standpoint, such MoE architectures successfully bound the active time complexity for the entire input sequence to $\mathcal{O}((N+M) \cdot D^2)$ (where $N$ and $M$ denote the sequence lengths of visual and textual tokens, and $D$ is the hidden dimension, as detailed in Table 1). However, this mathematical decoupling of parameters from FLOPs inherently risks representational fragmentation. Because tokens are discretely routed, the continuity of the model's internal feature space can be disrupted, occasionally leading to representation collapse if the routing distribution becomes highly skewed. Consequently, while in-domain performance scales efficiently, zero-shot generalization may degrade when confronted with complex multimodal inputs that lack a dense, unified pathway to synthesize orthogonal features.

## 2.4 Structural Decoding

Structural decoding enhances efficiency by architecturally constraining the interface between high-resolution perception and autoregressive generation. It builds on broader latent bottleneck and query-based sequence modeling ideas that predate modern multimodal models (Jaegle et al., 2021; Carion et al., 2020; Lee et al., 2019). Instead of allowing the language model to attend directly to dense, variable-length feature maps—which incurs prohibitive quadratic costs and modality misalignment—recent architectures introduce a learnable bottleneck that decouples decoding complexity from input dimensionality. Perceiver-style decoders like Flamingo (Alayrac et al., 2022) employ a resampler mechanism where fixed latent queries cross-attend to visual inputs, compressing spatiotemporal features into a constant number of visual tokens. Refining this principle, query-based transformers like BLIP-2 (Li et al., 2023b) utilize a lightweight Q-Former to act as a semantic bridge, actively distilling dense encoder outputs into a compact, text-aligned token set. Ultimately, structural decoding reframes efficiency as a problem of interface design: replacing exhaustive cross-attention with a bounded, fixed-capacity channel that effectively isolates the generative engine from the raw scale of sensory data while aligning heterogeneous modalities.

Mathematically, compressing a variable-length visual sequence $N$ into a fixed number of latent queries $K$ ($K \ll N$) drastically reduces the cross-modal interaction time complexity to $\mathcal{O}(K \cdot (N+M) \cdot D)$ (see Table 1). Yet, this architectural isolation inherently acts as an information bottleneck. This many-to-few mapping behaves analogously to a low-pass filter on the feature space, smoothing out high-frequency spatial details. Therefore, architectures relying on aggressive structural decoding face a clear representational trade-off: they excel at holistic semantic alignment (e.g., global VQA) but often struggle with tasks requiring dense,

Table 1: Comparison of Representative Multimodal Interaction and Adaptation Mechanisms. The empirical parameter counts are estimations based on canonical implementations. Theoretical complexity bounds reflect the core mechanism step, omitting auxiliary overheads. $N$ and $M$ denote the lengths of the token sequences for two distinct modalities, such as vision and language, respectively. $D$ is the hidden dimension, $K$ is the number of latent queries ($K \ll N$), $E$ is the number of experts, $r$ is the token retention ratio ($0 < r \leq 1$), and $r_{rank}$ is the intrinsic rank for parameter-efficient tuning ($r_{rank} \ll D$).

| Mechanism Category | Typical Params | Time Complexity | Space Complexity | Key Representational Trade-off |
|---|---|---|---|---|
| Unified Self-Attention (Baseline) | Backbone dependent | $\mathcal{O}((N+M)^2 \cdot D)$ | $\mathcal{O}((N+M)^2)$ | Unconstrained interaction; quadratic cost limits high-resolution |
| Dense Cross-Attention | $\sim$100M $-$ 1B | $\mathcal{O}(M \cdot N \cdot D)$ | $\mathcal{O}(M \cdot N)$ | High memory footprint; preserves exact feature alignment |
| Latent Query Bottleneck | $\sim$100M $-$ 200M | $\mathcal{O}(K \cdot (N+M) \cdot D)$ | $\mathcal{O}(K \cdot (N+M))$ | Many-to-few mapping limits fine-grained visual granularity |
| Linear Attention / SSMs | Varies (Arch. dependent) | $\mathcal{O}((N+M) \cdot D^2)$ | $\mathcal{O}(D^2)$ | Potential loss of precise token-to-token retrieval capability |
| Token Reduction (e.g., ToMe) | 0 (Training-free) | $\mathcal{O}((r \cdot N + M)^2 \cdot D)$ | $\mathcal{O}((r \cdot N + M)^2)$ | Analogous to low-pass filtering; smooths high-frequency details |
| Sparse Routing (MoE) | $\mathcal{O}(E \cdot D^2)$ total | $\mathcal{O}((N+M) \cdot D^2)$ | $\mathcal{O}(E \cdot D^2)$ | Risk of representation fragmentation and routing collapse |
| Modular Adaptation (e.g., LoRA) | $\sim$1M $-$ 50M | $\mathcal{O}((N+M) \cdot D \cdot r_{rank})$ | $\mathcal{O}(D \cdot r_{rank})$ | Rank restriction limits capacity for orthogonal multimodal shifts |

pixel-level grounding (e.g., small object detection), as the necessary representational granularity has been structurally pruned.

## 2.5 Modular Adaptation

Modular Adaptation mainly consists of bottleneck adapters and low-rank adaptation (LoRA) (Hu et al., 2022). These techniques originate from parameter-efficient adaptation methods for pretrained unimodal models and are later extended to support multimodal alignment and capability transfer (Houlsby et al., 2019). Adapters enhance efficiency by inserting lightweight, trainable modules between or within frozen pretrained components, enabling rapid specialization and cross-modal alignment without the cost of full-model retraining (Sung et al., 2022b;a; Pan et al., 2022; Sun et al., 2025b). Functionally, these architectures operate at two distinct structural levels: inter-module connection and intra-module tuning. For cross-modal alignment, projection adapters—such as the simple linear layers in LLaVA (Liu et al., 2023a) or the multi-layer perceptrons in MiniGPT-4 (Zhu et al., 2023)—act as minimal connectors that project sensory features directly into the language model's embedding space. Complementing these connectors, parameter-efficient tuning methods like LoRA (Hu et al., 2022; Guo et al., 2025b) introduce low-rank decompositions as bypass pathways inside transformer layers, allowing the model's internal reasoning to adapt to new modalities using a fraction of the original parameter count. Similar modular adaptation ideas also appear in speech-language and audio-language systems, where lightweight projection layers or adapter modules align acoustic representations with largely frozen language backbones, as in SALMONN (Tang et al., 2023a) and SALM (Chen et al., 2024e). Taken together, adapter-based strategies redefine efficiency as the structural decoupling of capability expansion from backbone maintenance—achieving scalable alignment and transfer under strict memory and compute budgets.

However, the mathematical constraints of modular adaptation impose distinct representational limits. As outlined in Table 1, methods like LoRA restrict the update space by an intrinsic rank $r_{rank}$ ($r_{rank} \ll D$), reducing the adaptation time complexity to $\mathcal{O}((N+M) \cdot D \cdot r_{rank})$. While this success is theoretically anchored in the intrinsic dimension hypothesis—which posits that models reside in a low-dimensional objective landscape during fine-tuning—this extreme low-rank constraint limits the model's capacity to internalize entirely new, orthogonal multimodal knowledge. Consequently, over-compressed adapters may struggle with complex cross-modal reasoning tasks that require mapping to entirely new feature spaces, occasionally leading to degraded generalization on out-of-distribution (OOD) domains.

## 2.6 Discussion and Key Insights

Model-level efficiency establishes the architectural foundation of EML by redesigning how computation is organized within and across modalities. Beyond the individual techniques cataloged above, our analysis reveals three fundamental paradigm shifts that define the next generation of efficient multimodal architectures:

- **From Explicit Perception to Latent-Core Abstraction:** The evolution from modality-specific pipelines to shared Transformer trunks and latent-core bottlenecks reflects a strategic shift toward

the information bottleneck principle. By forcing heterogeneous signals through a fixed-size latent array or a learnable resampler, architectures effectively decouple the quadratic cost of perception from the autoregressive complexity of reasoning. This structural constraint serves as a "semantic filter", ensuring that the generative engine processes only the most salient cross-modal alignments.

- **Addressing Representational Asymmetry via Sparsity:** Multimodal data exhibits inherent representational asymmetry in information density; for instance, visual tokens often contain significantly higher spatial and temporal redundancy compared to the dense semantics of text. Structural sparsity, particularly through MoE and conditional routing, enables architectures to address this asymmetry dynamically. Instead of a monolithic processing pass, modern models utilize modality-aware or content-driven routing to allocate high-capacity experts only to "hard" tokens while bypassing redundant patches.

- **Efficiency as a Structural Prerequisite Not a Trade-off:** A pivotal insight is that efficiency is starting to evolve from a post-hoc optimization into an intrinsic design primitive. Through mechanisms like modular adaptation and parameter-efficient tuning, efficiency is no longer viewed merely as a compromise on capability. Rather, it is a structural property that enables unprecedented model scalability—allowing systems to sustain effectively infinite context windows or perform real-time, omni-sensory reasoning that would be physically prohibitive under traditional dense architectures.

Ultimately, these model-level innovations culminate in a paradigm shift: the transition from modular concatenation toward natively unified foundations. By moving beyond the assembly of modality-specific encoders to form a cohesive world model, the architectural objective evolves from post-hoc alignment toward structural parsimony. Within this unified regime, efficiency is redefined not as a secondary trade-off or adjustment, but as an intrinsic, emergent property of the architecture's fundamental design, where computational capacity is natively and dynamically modulated by the latent semantic complexity of multimodal signals.

## 3 Algorithm

Algorithm-level efficiency defines *how* computation executes, compressing information flow within fixed architectural topologies. Unlike structural redesigns, these strategies target operation-level reductions in computation and memory footprint. As illustrated in Fig. 4, key techniques—ranging from token compression and quantization to state caching—systematically minimize data redundancy and arithmetic precision. By streamlining processing across both training-free and training-aware regimes, algorithm-level efficiency enhances runtime resource economy while preserving the model's structural integrity.

### 3.1 Token Compression & Selective Computing

Token compression in multimodal models builds on earlier token pruning and reduction methods developed in unimodal, especially in vision and long sequence modeling (Graves, 2016; Goyal et al., 2020). Reducing redundant visual tokens continues to be a central mechanism for accelerating multimodal transformers, where the computation cost grows quadratically with token count. The objective is not mere token removal, but to identify and retain semantically informative regions that drive multimodal understanding and reasoning. This area has evolved along two complementary lines—training-free and training-based compression—each reflecting a balance between practicality and adaptivity.

**Training-free Compression.** Training-free approaches perform pruning or merging during inference without altering pretrained parameters. While training-free methods inherit redundancy-based token reduction from unimodal models, their multimodal adaptation must ensure that discarded visual tokens do not remove evidence needed by the language model for grounding or fine-grained reasoning (Bolya et al., 2022). Early works exploit intrinsic attention maps or saliency patterns within vision encoders to prune tokens of low importance (Zhang et al., 2024c; Zhuang et al., 2025; Arif et al., 2025). Redundancy-aware methods extend this by measuring feature correlation or similarity to eliminate overlapping representations (Wen et al., 2025; Yang et al., 2025b; Tan et al., 2025). More recent advances adopt layer-adaptive or progressive pruning schedules that gradually reduce token counts across layers or decoding steps (Liu et al., 2024a; Zhuang

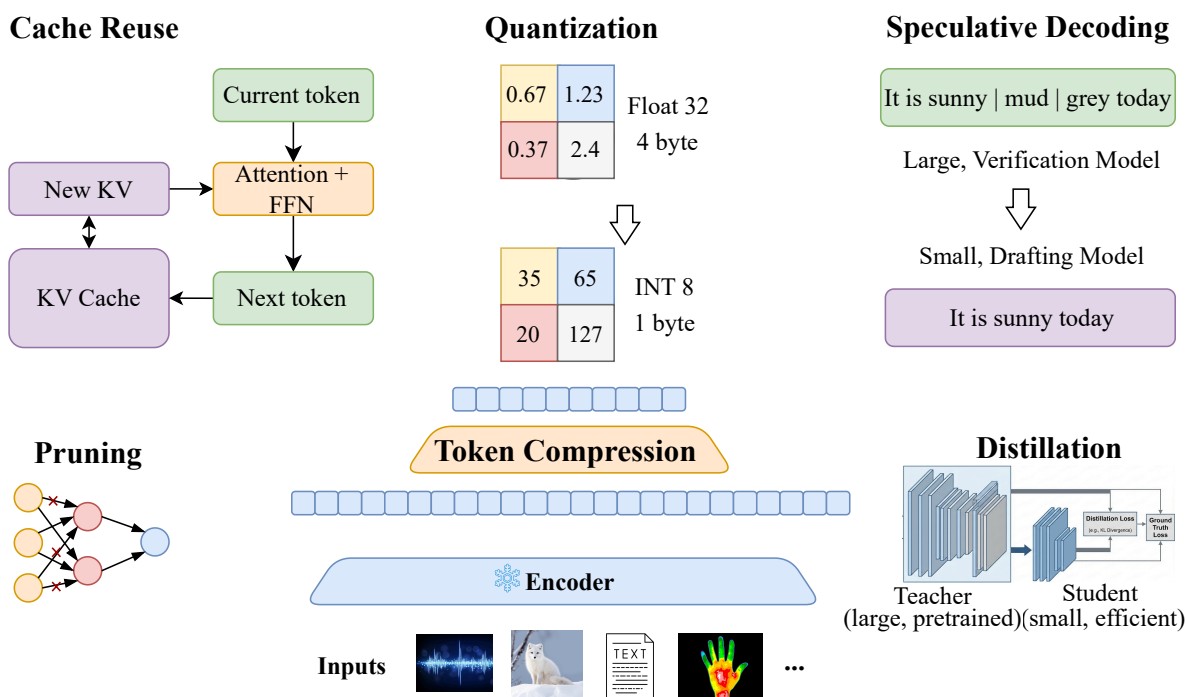

Figure 4: Algorithm-level efficiency for refining multimodal execution dynamics. This taxonomy illustrates the modulation of information flow across the EML pipeline through seven primary axes: (i) **Token compression and selective computing** to filter spatial redundancy and retain informative semantic regions; (ii) **Pruning** to eliminate structural redundancy within backbone architectures; (iii) **Quantization** to minimize memory bandwidth via precision discretization; (iv) **Knowledge distillation** to transfer reasoning behaviors and cognitive patterns to compact learners; (v) **Prompting and speculative decoding** to streamline input adaptation and parallelize generation; (vi) **Caching and reuse** to amortize prefill costs through temporal state persistence; and (vii) **Runtime sparsity** to enable adaptive computation based on input complexity. These strategies transform multimodal execution from static processing to a dynamic, information-flow-aware pipeline.

et al., 2025; Wang et al., 2025e), mitigating abrupt information loss. Structural and optimization-aware frameworks further refine this process by modeling token relationships as graphs (Jiang et al., 2025; Xing et al., 2024; Arif et al., 2025) or formulating selection as constrained optimization (Ye et al., 2025b; Omri et al., 2025), with recent omnimodal extensions such as OmniZip introducing training-free audio-guided audio-video token compression for unified multimodal inputs (Tao et al., 2025). Representative systems such as ST3 (Zhuang et al., 2025) and LLaVolta (Chen et al., 2024a) show that over 70% of visual tokens can be pruned while preserving multimodal accuracy, highlighting that meaningful compression is possible even without retraining.

**Training-based Compression.** In contrast to unimodal token reduction, training-based multimodal compression leans token selection jointly with downstream supervision, allowing the model to preserve tasks relevant cross-modal semantics rather than relying solely on intrinsic visual saliency (Rao et al., 2021; Yin et al., 2022). Learnable pruners such as P-Mod (Zhang et al., 2025a), FAST (Pertsch et al., 2025), and TokenPacker (Li et al., 2025d) introduce lightweight scoring networks or Q-Former structures that dynamically drop, aggregate, or compress high-rate acoustic and visual features based on a learnable importance $s_i$. Methods also compress audio encoder outputs before they are consumed by the LLM, combining token reduction with lightweight adaptation to recover downstream performance (Bhati et al., 2025). These models

are often trained with a task loss $\mathcal{L}_{\text{task}}$ with sparsity weight controlling factor $\lambda$:

$$\mathcal{L}_{\text{select}} = \mathcal{L}_{\text{task}} + \lambda \left\| \mathbf{s} \right\|_1, \tag{3}$$

where $\mathbf{s} = (s_1, s_2, ..., s_N), s_i \in [0, 1]$ denotes the vector of learnable importance scores, $N$ is the token length, $\lambda \left\| \mathbf{s} \right\|_1$ encourages sparsity in these scores. ViLA (Wang et al., 2024d) and Qwen-Audio (Chu et al., 2023) use a learnable text-guided sampling mechanism in the video and audio domains to selectively sample key frames or acoustic segments, demonstrating that joint optimization of modality selection and cross-modal distillation can yield significant speedups while improving accuracy. Pooling- and clustering-based frameworks (Wang et al., 2025b; Yang et al., 2025d) further abstract semantically similar tokens into compact latent embeddings, effectively reducing sequence length while preserving semantics. Curriculum-based approaches (Chen et al., 2024a) progressively decrease token budgets throughout training, encouraging models to adapt to increasingly compressed representations. By integrating token selection into optimization, these approaches achieve higher accuracy under heavy compression and enable controllable trade-offs between efficiency and semantic fidelity.

## 3.2 Pruning

Pruning (Han et al., 2015; Cheng et al., 2024) in multimodal models builds on broader neural network and compression literature, where redundant weights, layers or structured components are removed to reduce computation without extensive retraining. Early explorations often focused on unstructured pruning, which sparsifies individual weights based on magnitude or importance (LeCun et al., 1989; Zheng et al., 2025). While achieving high theoretical compression ratios, unstructured sparsity often fails to translate into actual speedup on standard hardware due to the irregular memory access patterns it creates.

To achieve tangible efficiency in multimodal learning, recent research has shifted toward Structured Pruning, which removes entire cohesive units such as neurons, attention heads, or layers. This approach aligns with hardware data layouts, enabling direct acceleration on CPUs and GPUs. The challenge in the multimodal context is to prune these structures while preserving delicate cross-modal alignments. Early multi-stage frameworks (Wang et al., 2024c) jointly prune visual and textual branches via layer-wise reduction (Sung et al., 2023b). Modern approaches like EfficientLLaVA (Liang et al., 2025a) formulate structured pruning as a generalization-aware search, automatically identifying redundant MLP blocks and attention heads using small proxy datasets.

Advancing beyond static reduction, UPop (Shi et al., 2023a) and MoPE-CLIP (Lin et al., 2024b) perform unified, progressive pruning across modalities. These methods dynamically allocate pruning ratios to different modal branches, ensuring that cross-modal representation remains balanced under high compression. Furthermore, MANU (Liu et al., 2025a) adopts modality-aware neuron pruning to eliminate cross-modal redundancies, improving both model compactness and representational disentanglement.

For pruning to be truly efficient in a system-level sense, it must transition from heuristic weight-trimming to hardware-friendly structural reduction. By targeting hardware-friendly components like attention heads and hidden dimensions, these methods ensure that the reduced model translates theoretical FLOPs savings into actual latency reduction, providing a scalable solution for deploying large-scale multimodal models on resource-constrained devices.

## 3.3 Quantization

Quantization compresses model footprint and accelerates inference by mapping high-precision floating-point weights and activations into lower-bit discrete representations (e.g., INT8) (Han et al., 2015; Jacob et al., 2018). Since modern large-scale multimodal models are frequently constrained by memory bandwidth as much as by raw arithmetic, reducing precision can improve cache utilization and enable more efficient low-precision execution on supported hardware. While unimodal quantization primarily focuses on preserving a single feature distribution, multimodal quantization must navigate the modality gap, the significant discrepancy in statistical ranges and outlier densities across visual, acoustic, and linguistic streams. If left unaddressed, uniform quantization can cause alignment drift, where the discretization noise applied to one modality disrupts the delicate semantic mapping to the language space, leading to a collapse in cross-modal

reasoning despite maintaining unimodal accuracy. Formally, uniform affine quantization maps a real-valued tensor $x$ to an integer $q$ via a scaling factor $\Delta$ and a zero-point $z$:

$$q = \text{clip}\left(\left\lfloor \frac{x}{\Delta} \right\rceil + z, \ q_{\min}, \ q_{\max}\right), \qquad \hat{x} = \Delta \left(q - z\right), \tag{4}$$

where $\lfloor \cdot \rceil$ denotes rounding to the nearest integer, $[q_{\min}, q_{\max}]$ defines the discrete range, and $\hat{x}$ represents the dequantized approximation.

**Post-Training Quantization (PTQ).** PTQ applies low-bit mapping directly to pretrained models without requiring extensive retraining, making it highly deployable. The underlying principles of PTQ, such as calibration, per-channel scaling and more, were first extensively developed in unimodal vision and language models (Ma et al., 2024b; 2023b; Banner et al., 2019; Nagel et al., 2019; Ma et al., 2024a; 2023a). However, multimodal models pose unique challenges due to the divergent statistical distributions of visual and textual features (Bhatnagar et al., 2025). Recent research addresses this via modality-aware calibration, where visual or audio tokens—which often exhibit distinct outlier patterns compared to text—receive differentiated scaling or grouping. Approaches like Q-VLM (Wang et al., 2024b), MBQ (Li et al., 2025c), VLM-Q (Xue et al., 2025), and MQuant (Yu et al., 2025) introduce sensitivity-based mixed precision, assigning higher bit-widths to varying channels or tokens that are critical for cross-modal alignment. Complementing weight-centric methods, STaMP (Federici et al., 2025) targets activation quantization, employing sequence transformations to suppress outlier activations that typically destabilize low-bit inference in large vision-language models.

**Quantization-Aware Training (QAT).** Aggressive compression regimes (e.g., sub-4-bit) often incur catastrophic discretization errors that PTQ fails to mitigate. QAT addresses this by simulating quantization-induced noise during the optimization trajectory, enabling the network to re-learn representational fidelity within a constrained bit-width (Hubara et al., 2018; Zhou et al., 2016). To overcome the prohibitive memory overhead of applying QAT to large-scale MLLMs, EfficientQAT (Chen et al., 2025b) introduces a tiered optimization paradigm: it utilizes block-wise training of all parameters followed by end-to-end refinement of quantization scales, effectively circumventing the "memory wall" while preserving the accuracy of 70B-scale models. In the context of multimodal reasoning, specialized challenges such as cross-modal outlier distributions are addressed by QSLAW (Xie et al., 2024), which integrates modality-aware warm-up and learnable step sizes into the instruction-tuning phase. QAT is also applied to the discretization of the input space itself. SpeechTokenizer (Zhang et al., 2023c) employs a learnable Residual Vector Quantization (RVQ) framework to hierarchically disentangle semantic content from acoustic details, ensuring the resulting speech tokens are optimized for language modeling rather than just signal reconstruction. Similarly, Retrieval-based Biasing (Flemotomos et al., 2025) employs vector quantization to approximate cross-attention scoring, enabling the efficient use of million-entry biasing catalogues by grounding retrieval in a discretized latent space. By co-optimizing discretization parameters with neural weights, these methodologies ensure that the delicate semantic alignment between modalities is preserved even in highly discretized spaces, bridging the gap between algorithmic compression and system-level inference throughput.

**Hardware Constraints.** In multimodal inference, the system must concurrently stream massive weights and high-fidelity sensory tokens (e.g., video or audio frames) from off-chip memory (DRAM/HBM) to the processor. By shrinking these operands from 16-bit to 4-bit, quantization effectively quadruples the functional bandwidth, allowing more information to reach the compute cores in a single clock cycle. However, this transition introduces hardware-level synchronization challenges: while modern GPUs and NPUs provide dedicated INT8 and INT4 Tensor Cores, they often struggle with mixed-precision misalignment. When various modality streams are quantized to different bit-widths to preserve alignment, the hardware may experience under-utilization of SIMD (Single Instruction, Multiple Data) units, as the processor must manage varying data layouts and dequantization overheads in real-time. Furthermore, the efficacy of quantization is strictly bound by the Native Instruction Set Architecture (ISA) of the underlying hardware. Even when modalities are unified under a single bit-width, the lack of hardware-level support for non-standard formats (e.g., 3-bit) can force the system into emulated execution. In these scenarios, the processor must perform real-time dequantization or use lookup tables to transform compressed operands into supported formats (like FP16) before computation (Lin et al., 2024c; Gholami et al., 2022).

### 3.4 Knowledge Distillation

Knowledge distillation (KD) has emerged as a pivotal technique for improving efficiency in multimodal learning by transferring knowledge from a large teacher to a smaller student model. Although many recent methods are developed for multimodal systems, their algorithmic foundations largely originate from unimodal distillation, including soft-target supervision for output alignment (Hinton et al., 2015), intermediate feature matching via hints (Romero et al., 2014), and structured representation transfer through attention- or relation-based objectives (Zagoruyko & Komodakis, 2016). More recent multimodal distillation also extends advances from language-model distillation, such as chain-of-thought and rationale transfer (Wang et al., 2023b). Building on these foundations, multimodal KD further incorporates modality-specific alignment, spatial grounding, and reasoning transfer across visual and textual streams. We group KD for multimodal systems into (i) prediction-level (logits/soft labels), (ii) representation-level (feature/attention/relational alignment), and (iii) behavior-level (reasoning traces, preferences, or policy signals).

**Prediction-level.** Prediction-level distillation transfers multimodal knowledge by aligning the output probability distributions of teacher and student models. It serves as the most implementation-friendly strategy, enabling efficiency gains without altering the student's internal structure. Formally, the objective minimizes the divergence between the teacher's soft logits $z_T$ and the student's logits $z_S$, often combined with ground-truth supervision:

$$\mathcal{L}_{\mathrm{KD}} = \alpha T^2 \, \mathrm{KL}\big(\sigma(z_T/T) \,\|\, \sigma(z_S/T)\big) + (1-\alpha) \, \mathrm{CE}\big(y, \, \sigma(z_S)\big), \tag{5}$$

where $\sigma$ denotes the softmax function, $T$ is the temperature parameter controlling distribution smoothness, and $\alpha$ balances the Kullback–Leibler (KL) divergence against the standard Cross-Entropy (CE) loss. While early approaches focus on simple logit matching for domain adaptation (Miech et al., 2021; Kang et al., 2025), recent works extend this to semantic grounding. PromptKD (Li et al., 2024e) employs prompt-based supervision to distill task priors without labeled data, while FDBPL (Zhang et al., 2025c) introduces region-aware binary prompts to transfer fine-grained spatial decision signals. In speech-centric multimodal settings, LiSER (Pendyala et al., 2025) distills emotion knowledge from both speech and visual teacher models into a lightweight speech student using confidence-aware softmax-level supervision over unlabeled audio-visual data. These methods provide a lightweight baseline for transferring teacher decisions through output imitation, often with minimal architectural modification.

**Representation-level.** Going beyond final predictions, representation-level distillation aligns intermediate features—such as attention maps, token embeddings, and relational matrices—to transfer the teacher's internal knowledge. Early works like CLIP-KD (Yang et al., 2024a) and TinyCLIP (Wu et al., 2023) validate direct feature matching to compress vision-language backbones. Recent advances target deeper structural alignment: DSMD (Liang et al., 2025b) employs dynamic scheduling to synchronize feature evolution, while SF-CLIP (Sameni et al., 2024) uses masked distillation to focus learning on salient spatial regions. Optimization also extends to architectural adaptation; MobileCLIP (Vasu et al., 2024) introduces dataset-level caching for efficient training, and mm-Mamba (Liao et al., 2025) utilizes progressive alignment to transfer Transformer-based knowledge into linear-time SSMs. Collectively, these strategies elevate KD from output mimicry to representational geometry transfer, preserving alignment fidelity under strict compute constraints.

**Behavior-level.** The most advanced form of KD transfers the teacher's underlying reasoning behaviors—how it decomposes problems, ranks alternatives, or formulates chains of thought (CoT). This paradigm captures decision dynamics rather than static snapshots, making it essential for complex instruction following. Systems such as Skywork R1V (Peng et al., 2025) introduce adaptive rationale-length supervision to balance completeness and conciseness in CoT generation, while VPD (Hu et al., 2024) employs visual–programmatic distillation to transfer explicit reasoning traces for structured tasks. Furthermore, preference-based methods like LLaVA-MoD (Shu et al., 2024) apply ranking distillation, where the student learns from the teacher's comparative judgments rather than full text reconstruction. Together, these methods mark a shift from predictive mimicry to cognitive emulation, allowing compact learners to approximate the deliberative, preference-driven reasoning of large VLMs.

### 3.5 Prompting and Speculative Decoding

This category of methods accelerates multimodal generation by optimizing the input conditioning logic rather than pruning the model structure. It encompasses two complementary strategies: learning compact prompts to adapt frozen backbones (reducing training/storage cost) and employing speculative drafting to parallelize decoding (reducing inference latency) (Leviathan et al., 2023; Lester et al., 2021).

**Parameter-Efficient Prompting.** Prompt learning turns adaptation into a continuous optimization problem over the input space. Early approaches treat prompts as static global adapters, injecting a small set of learnable tokens into frozen backbones to enable task transfer with negligible parameter cost (Jia et al., 2022; Li & Liang, 2021). Prefix Tuning (Li & Liang, 2021) formalizes this by attaching task-specific key–value prefixes at each Transformer layer, effectively steering the attention mechanism without modifying weights. Addressing the limitations of static prompts, CoCoOp (Zhou et al., 2022) introduces context- and instance-aware prompting, where the conditioning tokens evolve dynamically based on image features. This evolution from static to dynamic prompting allows models to achieve specialized performance with extreme parameter efficiency, often updating less than 1% of the total weights.

**Multimodal Speculative Acceleration.** Speculative decoding accelerates inference by breaking the memory-bound sequential dependency of autoregressive generation. It employs a lightweight draft model (or a prompt-conditioned head) to propose multiple tokens cheaply, which are then verified in parallel by the full target model (Yang et al., 2025a; Hu et al., 2025; Wang et al., 2025d). In multimodal contexts, efficiency is maximized by identifying shared resources: since the visual encoder is computationally heavy but static during decoding, systems share the visual KV cache between the drafter and the verifier. This allows the draft model to focus solely on linguistic prediction, creating a draft-and-verify loop that amortizes the high cost of loading the full model parameters over multiple accepted tokens per step.

### 3.6 Caching and Reuse

The Key-Value (KV) cache is a critical bottleneck in autoregressive generation, where memory consumption grows linearly with sequence length and batch size. Optimization strategies, many of which originated in unimodal models, aim to decouple memory growth from context length so that long-horizon generation remains feasible under fixed hardware budgets, while also accounting for the distinct characteristics of different modalities (Dai et al., 2019; Ma et al., 2026).

**Dynamic Eviction and Compression.** Importance-based methods mitigate cache redundancy by selectively identifying and discarding non-essential tokens (Zhang et al., 2023d; Liu et al., 2023c; Li et al., 2024d). Leveraging the high spatial redundancy of visual data relative to dense textual semantics, VL-Cache (Tu et al., 2024) introduces modality-aware pruning that aggressively evicts visual patches while shielding critical textual context. To enhance precision, ElasticCache (Liu et al., 2024c) and LOOK-M (Wan et al., 2024) employ attention entropy and anchor-based merging to differentiate between immutable semantic "anchors" and compressible repetitive patterns. Structural refinements, such as CSP (Pei et al., 2024), further stabilize this process by disentangling self-attention from cross-attention channels. On the deployment frontier, systems like FastCache (Zhu et al., 2025) and AirCache (Huang et al., 2025) implement retrieval-augmented hierarchies, offloading "cold" states to secondary storage while retaining "hot" states in GPU memory to balance extensive context horizons with limited hardware capacity.

**Temporal and Cross-Session Reuse.** Complementary to compression, reuse-oriented methods exploit the temporal coherence of multimodal signals to amortize computation. In dynamic scenarios like robotics or video streaming, the visual scene changes slowly. VLA-Cache (Xu et al., 2025) leverages this by reusing static background tokens across frames, recomputing only the dynamic patches relevant to the task. Similarly, Inf-MLLM (Ning et al., 2024) manages streaming inputs by identifying "attention saddle" points—tokens that sustain long-term dependencies—to maintain context over effectively infinite streams. Moving beyond single sessions, frameworks like MPIC (Zhao et al., 2025c) and Cache-of-Thought (Wu et al., 2025a) enable position-independent reuse. By projecting KV states into a transferable space or retrieving semantically related past states, these methods allow the model to reuse expensive prefill computations across different users or requests, significantly reducing latency for shared multimodal prompts.

### 3.7 Runtime Sparsity

Runtime sparsity optimizes efficiency by dynamically pruning the computation graph during inference based on input complexity. The underlying idea, including early exiting, depth-adaptive execution, and sparse token processing (Teerapittayanon et al., 2016; Elbayad et al., 2019; Rao et al., 2021), was first developed in unimodal adaptive inference and recent multimodal methods extend these principles to vision-language and embodied settings, where token importance and computational need vary substantially across modalities. Unlike static model pruning, which permanently removes parameters, runtime sparsity exploits the variance in sample difficulty—allocating full compute only to hard samples while processing easy ones via lightweight pathways. Structurally, these methods operate along two primary axes: reducing network depth (layer skipping or early exit) and sparsifying token connectivity (attention masking).

**Layer Skipping.** Layer skipping modulates computational depth by conditionally bypassing redundant Transformer blocks for easy tokens or frames. Observations suggest that many visual tokens stabilize early in the network, rendering deep processing unnecessary. Heuristic approaches like Skip-Vision (Zeng et al., 2025) prune low-impact visual tokens and their KV entries based on accumulated attention scores. Policy-based methods, such as D-ViTDMETR (Uzkent et al., 2023), employ reinforcement learning to make discrete execute-or-skip decisions per layer, reducing FLOPs by over 50% with minimal degradation. Recent advances treat depth as a continuous routing dimension; MoLe-VL (Zhang et al., 2025b) and $\lambda$-MoD (Luo et al., 2024) learn sparse activation paths based on token entropy or spatiotemporal salience. Similarly, mPLUG (Li et al., 2022) utilizes structural shortcuts to enable speculative partial-depth execution, demonstrating that adaptive depth can effectively balance representation power with inference speed.

**Early Exit and Adaptive Termination.** Early exit mechanisms transform fixed-depth backbones into dynamic cascades, allowing inference to terminate at intermediate layers once a confidence threshold is met. In vision-language tasks, frameworks like DeeCap (Fei et al., 2022) demonstrate that shallow layers often contain sufficient semantic information for simple captioning instances. Similarly, HuBERT-EE (Yoon et al., 2024) adapts this to acoustic modeling by attaching auxiliary predictors to intermediate layers of a HuBERT backbone. More rigorously, MuE (Tang et al., 2023b) formalizes this via convergence monitoring, halting computation when cross-modal representations saturate. In temporal or embodied contexts, efficiency is driven by stability; DeeR (Yue et al., 2024) halts processing for video frames or robotic actions once the policy distribution stabilizes over time. While some solutions like AdaInfer (Fan et al., 2024) rely on parameter-free statistical checks, others like CREMA (Yu et al., 2024) and CAPEEN (Bajpai & Hanawal, 2024) integrate exit decisions into the training loop, distilling knowledge from deep layers to shallow exits to ensure consistent performance regardless of termination depth.

**Attention Sparsity and Merging.** Attention sparsity mitigates the quadratic complexity of self-attention by restricting connection density. Static methods like ETESparse (Dai et al., 2021a) utilize fixed-pattern constraints, while content-adaptive approaches such as LoRA-Sparse (Song et al., 2024) dynamically compute attention only over top-ranked keys to minimize redundant computation. Complementing these structural reductions, merging paradigms offer a training-free pathway for efficiency through representation aggregation. At the model level, TIES-Merging (Yadav et al., 2023) and related benchmarks (Sung et al., 2023a) consolidate diverse parameter sets from multiple checkpoints to enhance performance without adding inference overhead. At the token level, LLaVA-PruMerge (Shang et al., 2025) introduces an adaptive framework that synergistically combines cross-modal attention-based pruning with similarity-driven merging. By dynamically identifying task-relevant visual tokens and aggregating redundant features, it significantly compresses the input space for the LLM while maintaining high-fidelity multimodal reasoning. Together, these methods optimize connectivity by either pruning non-essential interactions or aggregating similar representations into dense, informative states.

### 3.8 Discussion and Key Insights

Algorithm-level efficiency optimizes the execution dynamics of multimodal models by actively modulating the density and precision of information flow. Our synthesis of recent advancements reveals three critical insights into how algorithmic strategies move beyond post-hoc compression toward intelligent, adaptive computation:

- **Exploiting Asymmetric Redundancy for Token Economy:** A core pillar of algorithmic efficiency is the identification of asymmetric redundancy across modalities. While textual tokens are semantically dense and sequential, tokens from other modalities—such as visual and audio streams—often exhibit high spatial and temporal correlation. Effective token compression and selective computing (Wen et al., 2025) succeed by treating visual patches not as independent units, but as a hierarchical graph of information. This allows models to achieve high pruning rates by focusing computational budgets on "high-entropy" regions while aggressively aggregating static background tokens.

- **Sensitivity-Aware Discretization in Heterogeneous Spaces:** The primary challenge in multimodal quantization lies in the divergent statistical distributions of visual and textual features. Our analysis suggests that the most robust quantization strategies—such as sensitivity-based mixed precision and outlier suppression—succeed by acknowledging this heterogeneity (Xue et al., 2025). Rather than applying a uniform bit-width, these methods protect the delicate cross-modal alignment by maintaining higher precision in channels or tokens that act as "semantic anchors", while quantizing redundant activations to ultra-low-bit levels.

- **From Predictive Mimicry to Cognitive Emulation:** The paradigm shift in KD from simple logit-matching to behavior-level emulation (Peng et al., 2025) highlights a deeper goal: transferring the reasoning process itself. By distilling reasoning traces, preference signals, and chain-of-thought (CoT) behaviors, algorithmic efficiency enables compact student models to approximate the deliberative reasoning of large-scale teachers. This transforms efficiency from a reduction of FLOPs into a maximization of cognitive throughput.

Ultimately, the trajectory of algorithm-level innovation heralds a decisive paradigm shift toward cognitive-aware dynamic orchestration. By transcending the rigid constraints of static execution through the strategic synergy of runtime sparsity and speculative decoding, next-generation EML frameworks are poised to resolve the fundamental "Efficiency-Utility-Alignment" trilemma. Within this nascent regime, algorithms will function as elastic reasoning engines, autonomously and natively modulating computational expenditure in direct response to the latent semantic complexity of the multimodal stream.

## 4 System

While the Model and Algorithm layers of EML have experienced explosive growth and reached a degree of methodological maturity, **multimodal-specific system orchestration remains in its nascent stages.** The current research landscape is heavily skewed toward architectural design and weight compression. However, as multimodal models transition from laboratory benchmarks to real-time, edge-cloud deployments, the bottleneck fundamentally shifts from theoretical FLOPs to physical constraints such as KV cache limits, I/O bandwidth, and heterogeneous scheduling.

Distinct from model and algorithm optimizations, system-level approaches focus on resource orchestration—determining *where* and *when* workloads should execute under these physical constraints. This layer operationalizes efficiency by balancing infrastructure trade-offs between latency, memory, and energy, as shown in Fig. 5. In this section, we synthesize this highly emerging frontier, examining critical strategies for scalable deployment: context-aware memory management, edge–cloud collaboration, latency-sensitive scheduling, and hardware–software co-design.

### 4.1 KV Cache Management and Serving

In production environments, KV cache management is the primary determinant of system throughput and maximum concurrent users (Kwon et al., 2023). Profiling studies by Lee et al. (Lee et al., 2025b) reveal that for long-context workloads, end-to-end latency is governed by memory bandwidth and kernel launch overheads rather than arithmetic intensity. To address this, modern serving engines integrate `torch.compile`, CUDA graph execution, and FlashAttention to fuse kernels and minimize scheduling gaps. Building on this, research focuses on orchestrating the lifecycle of KV states to maximize hardware utilization.

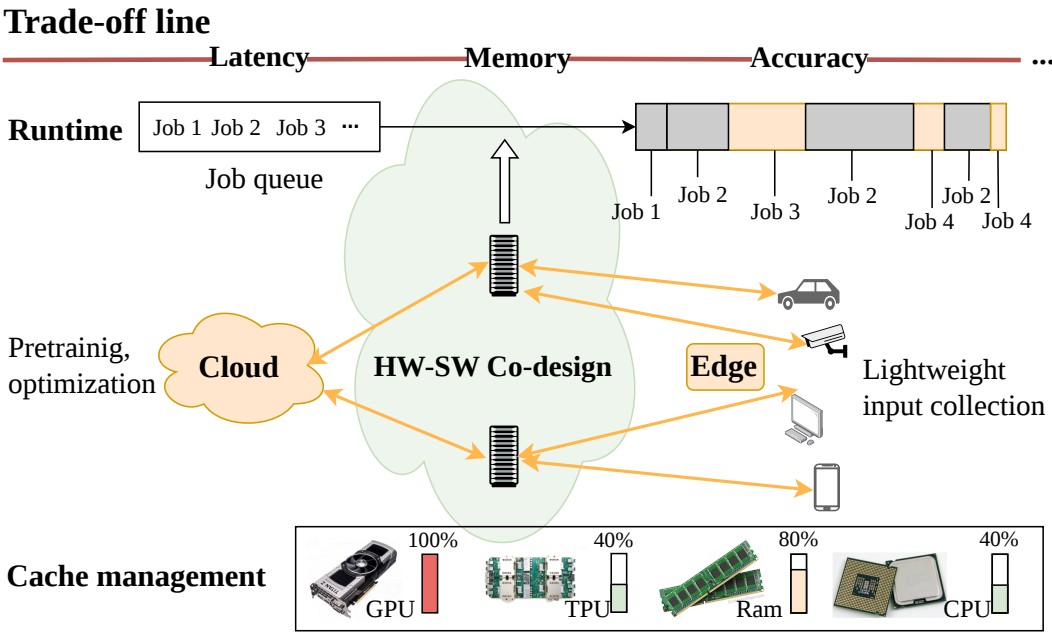

Figure 5: System-level efficiency for elastic resource orchestration. This framework illustrates the final operationalization of EML, where theoretical gains are translated into realized performance across five primary axes defined in our MAS taxonomy: (i) **KV Cache Management and Serving** to decouple memory growth from sequence length and optimize throughput; (ii) **Edge-cloud Collaboration** for establishing hierarchical cognitive pipelines and uncertainty-guided offloading; (iii) **Latency-Aware Scheduling and Pipelining** to maximize hardware utilization by reordering and overlapping cross-modal requests; and (iv) **Hardware-software (HW-SW) Co-design** to natively align model architectural topology with the physical constraints of heterogeneous accelerators. Collectively, these strategies transform static multimodal execution into a dynamic, hardware-aware ecosystem.

**High-Throughput Memory Pooling.** Instead of viewing compression merely as a way to reduce FLOPs, system-level approaches treat it as a mechanism to increase batch size and GPU occupancy. FastCache (Zhu et al., 2025) introduces a stream-aware memory pool that dynamically manages KV blocks across concurrent decoding sessions. By optimizing the physical layout of cached states, it mitigates memory fragmentation—a common issue in dynamic length generation—thereby removing I/O bottlenecks to support high-throughput serving. Similarly, MEDA (Wan et al., 2025) utilizes per-layer entropy to guide adaptive memory allocation, ensuring that scarce VRAM is physically reserved for information-dense layers, increasing the maximum supported sequence length on a single device.

**Shared State Serving and Persistence.** For multi-turn or multi-user scenarios, the system bottleneck shifts to the repeated loading of redundant contexts. Reuse-oriented architectures decouple the logical KV state from its physical storage, enabling zero-copy sharing across requests. MPIC (Zhao et al., 2025c) implements position-independent caching, allowing the serving backend to map multiple user prompts to the same shared physical memory block, regardless of their position indices. This significantly reduces the memory bandwidth requirement for prefix loading. Building on the concept of persistent memory, video-SALMONN S (Sun et al., 2025a) introduces a Test-Time Training (TTT) memory module. Unlike standard KV caches that grow linearly, this approach continually updates token representations via a Hessian-free conjugate-gradient procedure ($TTT_{HF}$), effectively embedding three hours of video history into a fixed-size parameter state. At the industrial scale, Gemini 1.5 (Team et al., 2024) demonstrates the necessity of unified state management, where a centralized cache orchestration layer enables persistent context reuse across massive, heterogeneous multimodal streams. This transition from per-request computation to persistent shared memory transforms the KV cache from a temporary buffer into a globally managed system asset.

## 4.2 Edge-Cloud Collaboration

Real-world multimodal deployment operates under a fundamental tension: edge devices suffer from compute limitations, while cloud servers face latency and bandwidth bottlenecks. Edge–cloud collaboration resolves this by establishing a hierarchical computing architecture that dynamically partitions workloads based on resource availability and privacy concerns (Kang et al., 2017).

**Adaptive Inference Offloading.** This paradigm treats the edge as a low-latency semantic filter and the cloud as a high-capacity reasoning engine. System-level frameworks like EdgeCloudAI (Ghasemi et al., 2024) and MoA (Yang et al., 2025f) implement split-computing architectures. Instead of transmitting raw video streams, lightweight edge modules (e.g., CNNs or small VLMs) perform preliminary analytics to filter redundant frames or extract feature embeddings. For instance, Zhu et al. (Zhu et al., 2026) utilize a teacher-guided difficulty classifier to predict the optimal early-exit (EE) layer for speech translation. Simple utterances are processed at shallow layers on the edge, while complex samples are offloaded to the cloud. Evaluations on testbeds like NSF COSMOS (Raychaudhuri et al., 2020) demonstrate that this adaptive partitioning significantly reduces bandwidth consumption and end-to-end latency while maintaining near-cloud accuracy.

**Collaborative Continuous Learning.** Beyond static inference, system efficiency also entails maintaining model relevance over time without incurring massive data transfer costs. Frameworks for Cloud–Device Collaborative Adaptation (Gan et al., 2023) introduce uncertainty-guided interaction mechanisms. Here, edge devices utilize uncertainty quantification to identify out-of-distribution or ambiguous samples. Only these "hard" examples are uploaded for cloud-based labeling and training, after which a distilled, lightweight update is synchronized back to the edge. This bidirectional loop minimizes communication overhead and preserves privacy by keeping routine data local, ensuring that the system continuously adapts to evolving environments with minimal operational cost.

## 4.3 Latency-Aware Scheduling and Pipelining

Achieving low-latency multimodal inference requires moving beyond simple sequential execution to sophisticated scheduling that maximizes hardware utilization (Yu et al., 2022a). Multimodal models introduce a unique system challenge: the "stalled pipeline" problem, where the large language model (LLM) often sits idle waiting for other modalities' encoders to process high-resolution images. System-level scheduling addresses this by optimizing the timeline of execution—reordering, overlapping, and routing requests to hide latency bubbles.

**Asynchronous Pipelining and Overlap.** To alleviate the computational stalls inherent in sequential cross-modal dependencies, recent system-level frameworks prioritize fine-grained temporal parallelism. RServe (Guo et al., 2025a) introduces a split-scheduling architecture that actively interleaves compute-intensive vision encoding with the language prefill phases of concurrent requests. By decoupling these execution stages and managing per-request embeddings asynchronously, it effectively masks the latency of visual encoding, yielding a 2× throughput improvement over traditional sequential serving. For high-throughput streaming environments, Inf-MLLM (Ning et al., 2024) employs a streaming-aware scheduler that dynamically governs the token memory lifecycle through adaptive KV retention. This methodology enables sustained long-context generation on a single GPU while significantly curtailing the latency spikes characteristic of heuristic eviction policies such as H2O (Zhang et al., 2023d).

**Heterogeneous Routing and Tiering.** In distributed environments, latency is optimized by routing requests to the most appropriate resource tier. Yuan et al. (Yuan et al., 2025) propose a decoupled architecture for edge-cloud systems, where lightweight encoders run on edge devices and LLMs on servers. They employ a Gaussian Process-Upper Confidence Bound (GP-UCB) scheduler to dynamically select the optimal offloading target and power configuration, minimizing energy consumption under strict latency constraints. At the production scale, Amazon Nova (Langford et al., 2025) adopts a tiered deployment strategy (Micro, Lite, Pro, Premier). By routing queries based on complexity—sending simple captioning tasks to cheaper models and complex reasoning to larger ones—the system satisfies strict Service Level Agreements (SLAs) while optimizing the global cost-latency trade-off.

### 4.4 Hardware-Software Co-design

As multimodal models scale, system efficiency is increasingly constrained by the mismatch between algorithmic requirements and hardware capabilities. Specifically, multimodal workloads exhibit diverse *arithmetic intensities*: dense layers in LLMs are typically compute-bound, while high-resolution visual encoders or attention mechanisms are often memory-bound. Hardware-software co-design addresses this by aligning the model's operational structure with the physical topology of the underlying accelerators (Jouppi et al., 2017).

**Modality-Aware Heterogeneous Mapping.** A core challenge in EML is the diverse arithmetic intensities across modalities: dense LLM layers are typically compute-bound, while high-resolution vision encoders or cross-attention mechanisms are often memory-bound. Hardware-software co-design addresses this by dynamically mapping modality-specific kernels onto the most suitable hardware units (Jouppi et al., 2017). Golden et al. (Golden et al., 2024) demonstrate that treating all modal operators uniformly leads to resource underutilization. By characterizing the unique compute-to-memory ratios of vision, audio, and text kernels, they propose a heterogeneity-aware mapping strategy. Compute-dense modules are routed to high-FLOP units (e.g., systolic arrays in GPUs/TPUs), while bandwidth-bound operations are allocated to memory-rich processors or near-memory compute units. This approach minimizes data movement—the primary energy consumer in modern infrastructure—by ensuring a structural-temporal alignment between the model topology and the physical accelerator.

**Joint Architecture-Hardware Search.** Instead of fitting a fixed model onto hardware, the second strategy uses Neural Architecture Search (NAS) to co-optimize the model structure and its deployment parameters. Harmonic-NAS (Ghebriout et al., 2024) exemplifies this joint optimization paradigm. It searches for modality-specific backbones and fusion networks under strict hardware constraints (e.g., latency, energy, or peak memory on an edge device). By incorporating hardware feedback directly into the search loop, it discovers a Pareto frontier of architectures that maximize accuracy within specific physical budgets. This effectively shifts the design process from a sequential "train-then-deploy" workflow to a unified co-design loop, producing hybrid architectures natively efficient for their target deployment platforms.

### 4.5 Discussion and Key Insights

System-level efficiency marks the final operationalization of EML, where theoretical algorithmic gains are translated into realized performance under physical constraints. Our synthesis of memory pooling, edge-cloud partitioning, and HW-SW co-design reveals that efficiency has evolved from localized operator optimization into a global orchestration of data movement and hardware-native adaptability:

- **From Arithmetic Intensity to I/O-Bound Orchestration:** As evidenced by the shift toward high-throughput memory pooling and shared-state serving, the primary bottleneck in MLLM deployment has migrated from compute cycles to memory bandwidth. True system efficiency now resides in the fluidity of the KV cache lifecycle. By treating the KV cache as a globally managed, persistent asset rather than a transient buffer, systems can decouple memory growth from context length, effectively transforming memory fragmentation into a manageable pool of high-speed tokens.

- **Hierarchical Cognitive Pipelining via Edge-Cloud Symbiosis:** The integration of adaptive offloading transcends simple workload partitioning; it establishes a hierarchical pipeline mirroring biological sensory systems. By utilizing the edge as a low-latency semantic filter to prune modal redundancy, the system reserves cloud-scale reasoning for high-value outliers. This synergy ensures that efficiency is maintained through uncertainty-guided adaptation across the device-cloud continuum, balancing the trilemma of latency, bandwidth, and accuracy.

- **Structural-Temporal Symbiosis as a Design Necessity:** The "stalled pipeline" problem identified in multimodal workloads exposes the mismatch between fixed hardware topologies and dynamic cross-modal dependencies. Solutions like RServe (Guo et al., 2025a) demonstrate that efficiency emerges when the model's structural topology is natively aligned with its temporal execution. Through joint HW-SW co-design, the development process evolves into a unified loop that elimi-

nates pipeline bubbles and minimizes data movement—the primary energy consumer in modern infrastructure.

- **Heterogeneous Mapping and Modality-Aware Scheduling:** A critical insight from recent characterization studies is that multimodal efficiency is a function of hardware-modality alignment. Since vision, audio, and language kernels exhibit vastly different computational densities, a monolithic execution strategy is inherently inefficient. The future of EML systems hinges on modality-aware autonomous scheduling that dynamically dispatches diverse kernels onto specialized heterogeneous compute units (e.g., NPUs for visual encoding vs. GPUs for LLM reasoning), thereby striking a Pareto-optimal balance between energy efficiency and operational throughput.

Ultimately, these insights posit that the future of EML lies in **elastic resource orchestration**—a self-regulating infrastructure that dynamically reconfigures its compute, memory, and communication pathways to satisfy the volatile, multi-objective demands of real-world multimodal intelligence.

## 5 Efficient MLLMs

Rather than an isolated research niche, Efficient MLLMs serve as the ultimate proving ground for our MAS framework. The evolution of this field, as visualized in Fig. 6, reveals a distinct maturity curve: a trajectory that begins with **architectural consolidation**, pivots toward **algorithmic refinement**, and is currently converging on **system-level orchestration**.

### 5.1 The Evolutionary Trajectory: From Models to Systems

The chronological progression of MLLM efficiency (see Fig. 6) reflects the shifting bottlenecks of large-scale efficient multimodal intelligence:

- **The Foundational Era (Model-Centric):** Initial efforts primarily targeted the training bottleneck. Early frameworks like BLIP-2 (Li et al., 2023b) and LLaVA (Liu et al., 2023a) focused on minimizing the cost of cross-modal alignment through modular adapters and frozen backbones. This era established the principle of "architectural frugality", where efficiency was synonymous with parameter-efficient fine-tuning.

- **The Inference Pivot (Algorithm-Centric):** As MLLMs moved toward deployment, the bottleneck shifted to memory and latency limits. Algorithm-level strategies matured from static pruning to dynamic runtime reduction. Innovations like LOOK-M (Wan et al., 2024) and Spec-VLA (Wang et al., 2025d) demonstrate a move toward "semantic-aware execution", where computation is selectively allocated to critical tokens (e.g., text-guided KV merging) to sustain long-context reasoning within fixed hardware envelopes.

- **The Deployment Frontier (System-Centric):** Most recently, we observe a significant surge in system-level optimizations—a trend driven by the transition from lab prototypes to high-throughput production. Works such as MiniCPM-V (Yao et al., 2024), EdgeShard (Zhang et al., 2024b), MPIC (Zhao et al., 2025c), and the speed-oriented co-design of Gemini 3-Flash (Google, 2025) highlight that the current frontier lies in resource orchestration. This shift acknowledges that even the most efficient model can fail in real-world scenarios without asynchronous pipelining and elastic memory management.

### 5.2 Discussion: Vertical Synergy as the Final Frontier

Our analysis of the MLLM landscape reveals three key insights regarding the synergy between MAS layers:

- **Decoupling Capacity from Cost:** Structural sparsity (e.g., MoE-LLaVA (Lin et al., 2024a)) represents a successful synthesis of model-level topology and algorithmic routing. By decoupling total parameters from active FLOPs, MLLMs are evolving into "sparse-active" systems that can scale in knowledge without scaling in inference latency.

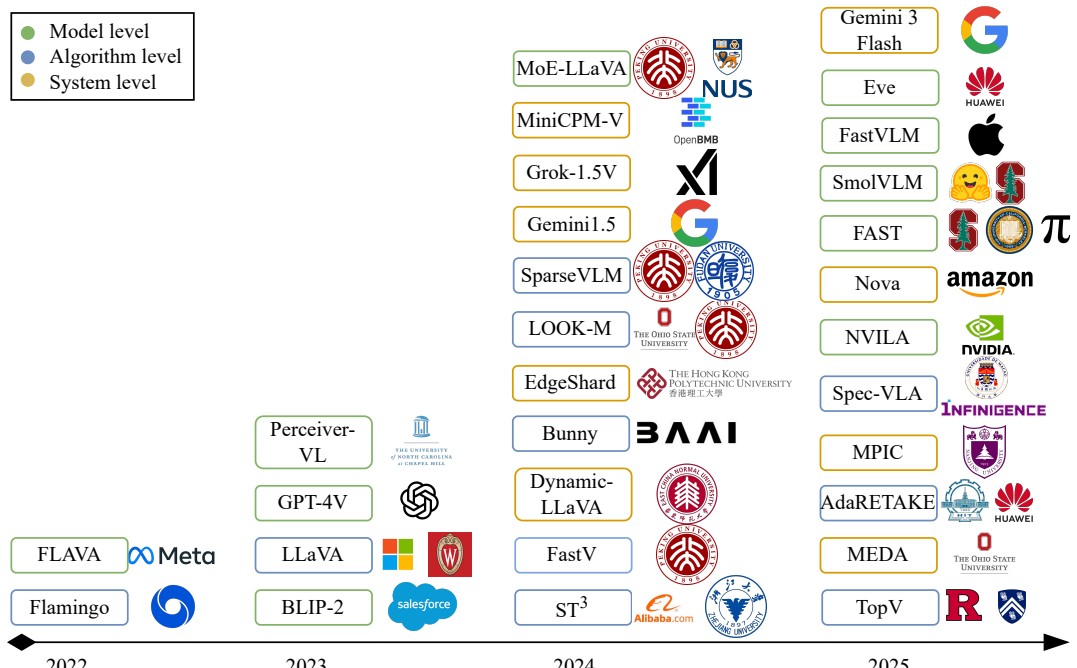

Figure 6: Chronological overview of representative efficient MLLMs. Models are categorized by primary optimization level: **model-level** (green), **algorithm-level** (blue), and **system-level** (orange). This distribution highlights a distinct paradigm shift, where model- and algorithm-level optimizations are dominant in the early stages, while system-level resource orchestration has gained significant prominence in recent years.

- **The Convergence of Perception and Scheduling:** Recent systems like FastVLM (Vasu et al., 2025) and NVILA (Liu et al., 2025b) suggest that the boundary between "visual perception" and "system scheduling" is blurring. By designing architectures that are natively compatible with CUDA-graph execution and kernel fusion, these models ensure that architectural efficiency translates directly into wall-clock speed gains.

- **Towards Self-Regulating Computation:** The emerging trend of adaptive efficiency (e.g., MiniCPM-V (Yao et al., 2024)) suggests a future where MLLMs learn not only to reason, but to *self-regulate*. In this paradigm, efficiency becomes an internal optimization objective where the model autonomously decides the optimal path across the MAS stack—choosing *what* (Model), *how* (Algorithm), and *where* (System) to compute based on real-time constraints.

To conclude, efficient MLLMs serve as the genesis of multimodal systems where efficiency is no longer an afterthought, but an intrinsic, self-regulating property of intelligent computation.

# 6 Applications and Benchmarks

After detailing the methodological advances in the preceding sections, we now turn to the diverse application domains and benchmarking ecosystems that serve as the primary drivers for EML. As summarized in Table 2, these domains are not merely downstream tasks but represent distinct frontiers where the efficiency-performance trade-off is governed by specific physical and operational constraints. From the millisecond-latency requirements of autonomous driving to the memory-intensive horizons of long-video understanding, these applications necessitate a tight integration of MAS optimizations.

Table 2: A Taxonomy of application domains and benchmarks in EML. This table highlights the diverse modality combinations—ranging from canonical audio-visual pairs to complex sensor streams—that necessitate domain-specific MAS optimization strategies to balance computational fidelity with deployment constraints. V: Visual, T: Text, A: Audio, S: Spatial (LiDAR+GPS+Radar), D: Depth, Act: Action, Sen: Sensor, Tab: Tables, Cha: Charts.

| Applications | Task | Benchmark | Modalities |
|---|---|---|---|
| Affective Computing | Sentiment Analysis, Emotion & Behavior Recognition | CMU-MOSI (Zadeh et al., 2016), CMU-MOSEI (Zadeh et al., 2018), IEMOCAP (Busso et al., 2008), MELD (Poria et al., 2019), CH-SIMS (Yu et al., 2020) | V+T+A |
| Embodied AI & Robotics | VLN, Manipulation, Embodied Agents | R2R (Anderson et al., 2018), REVERIE (Qi et al., 2020), AL-FRED (Shridhar et al., 2020), Ego4D (Grauman et al., 2022), TEACh (Padmakumar et al., 2022), Habitat (Savva et al., 2019) | V+D, T+S+Act |
| Media Understanding & Generation | Video QA, Streaming Analysis, Retrieval & Generation | MSR-VTT (Xu et al., 2016), TVR (Lei et al., 2020b), Video-MME (Fu et al., 2024), MVBench (Li et al., 2024b), LAION-5B (Schuhmann et al., 2022), WebVid2M (Bain et al., 2021), TVQA+ (Lei et al., 2020a), AVA ActiveSpeaker (Roth et al., 2020) | V+T+A |
| Healthcare | Radiology Analysis, Clinical Speech, Digital Twin | MIMIC-CXR (Johnson et al., 2019), IU-Xray (Demner-Fushman et al., 2015), PPMI (Marek et al., 2011), MedVidQA (Gupta et al., 2023), mPower (Bot et al., 2016) | V+T+A+Sen |
| Spatial Understanding | Autonomous Driving, 3D Scene Reconstruction | nuScenes (Caesar et al., 2020), Waymo Open Dataset (Sun et al., 2020), KITTI (Geiger et al., 2012), Argoverse (Chang et al., 2019), Drive&Act (Martin et al., 2019) | V+S+IMU |
| Multimodal Reasoning | VQA, Math Reasoning, Visual Dialogue | VQAv2 (Goyal et al., 2017), GQA (Hudson & Manning, 2019), OK-VQA (Marino et al., 2019), VCR (Zellers et al., 2019), ScienceQA (Lu et al., 2022), MathVista (Lu et al., 2024b), ChartQA (Masry et al., 2022) | Tab+Cha+V+T |
| Federated Learning | Privacy-Preserving Collaborative Training, Cross-Device Alignment | FedMultimodal (Feng et al., 2023), Med-MMFL (Chhetri et al., 2026) | V+T+A+Sen |

## 6.1 Affective Computing

Affective computing serves as a critical frontier for EML, where the objective of inferring nuanced human emotions from tri-modal cues—audio, visual, and linguistic—is fundamentally constrained by temporal transience and stringent privacy imperatives (Wang et al., 2025c; Sun et al., 2026; Lian et al., 2024b;a). The core efficiency challenge in this domain lies in the high-frequency and ephemeral nature of affective signals, which necessitates a transition from computationally expensive monolithic encoders toward modular, on-device adaptation. Within our MAS framework, affective computing exemplifies a classic case of cross-layer

synergy: model-level adapter tuning minimizes the memory footprint of backbone networks to enable rapid task specialization, while algorithm-level innovations—such as the dynamic token pruning and quantized inference utilized in UGotMe (Li et al., 2025b)—exploit the inherent spatiotemporal redundancy in facial and vocal features to achieve real-time responsiveness. This structural decoupling ensures that the generative reasoning engine is only invoked for high-entropy emotional transitions, while routine behavioral monitoring is offloaded to efficient primitives. Ultimately, these advancements move toward a paradigm of empathetic edge intelligence, where system-level constraints on latency and power consumption dictate the selection of lightweight fusion strategies, ensuring that affective intelligence remains continuous, responsive, and privacy-preserving without reliance on cloud-based orchestration.

## 6.2 Embodied AI and Robotics

Embodied AI and robotics represent the physical manifestation of EML, where the sensory-motor loop demands absolute synchronization between high-dimensional perception and real-time action generation under strict latency envelopes (Han et al., 2026). The primary efficiency bottleneck in this domain arises from the explosive computational complexity of Vision-Language-Action (VLA) trajectories, necessitating a transition from static mapping to resource-aware execution (Chen et al., 2025c). Within our MAS framework, foundational systems like PaLM-E (Li et al., 2025e) and RT-2 (Zitkovich et al., 2023) exemplify how model-level structural priors and system-level asynchronous pipelining coalesce to hide the latency of compute-intensive visual encoding behind lightweight action decoding loops. This evolution highlights a broader movement toward adaptive sensor orchestration, where the agent dynamically modulates its perceptual resolution or sensor sampling rate based on task uncertainty. By utilizing the MAS system layer to prioritize computational budgets for high-stakes manipulation or navigation sub-goals, these systems effectively bridge the gap between abstract understanding and embodied reality, ensuring that multimodal intelligence remains responsive and scalable within the physical bounds of mobile hardware.

## 6.3 Media Understanding and Generation

Media understanding and generation drive the frontiers of EML by interpreting and synthesizing content across vast spatiotemporal horizons, where efficiency is governed by the quadratic complexity of multimodal attention (Ye et al., 2025a; He et al., 2024a; Song et al., 2026; Lin et al., 2026). The core challenge in this domain is to exploit the massive inherent redundancy in high-resolution video and audio streams to enable long-context reasoning without memory collapse. Recent innovations pivot from dense global modeling toward content-aware execution, utilizing algorithmic token pruning and temporal windowing—as demonstrated in Video-ChatGPT (Maaz et al., 2024) and CLIPER (Wu et al., 2024b)—to linearize the computational footprint of high-resolution media. This domain serves as a primary proving ground for MAS system-level orchestration, where the reuse of KV cache states and the dynamic offloading of cold tokens allow MLLMs to sustain effectively infinite context windows for streaming analysis. By aligning model-level structural sparsity with system-level memory persistence, media systems are evolving into responsive, long-form interpreters capable of real-time retrieval and generative QA across massive, heterogeneous archives.

## 6.4 Healthcare

In clinical intelligence, efficiency is a structural necessity due to massive imaging data, strict privacy regulations, and limited on-site compute. The goal is to maximize diagnostic fidelity while minimizing the computational footprint of multimodal stacks. Recent advances demonstrate a shift toward model-level specialization (Zhao et al., 2026); for instance, pretraining frameworks like BioViL-T (Bannur et al., 2023) extend beyond static image-report pairs to model longitudinal radiology studies, improving report grounding through the structural alignment of temporal data. Complementing this, algorithm-level innovations focus on augmenting compact backbones with structured expertise. Knowledge-enhanced systems like KARGEN (Li et al., 2024c) fuse disease graphs with frozen LLMs to mitigate hallucinations, proving that specialized, instruction-tuned architectures like CXR-LLaVA (Lee et al., 2025a) can outperform massive general-purpose VLMs in radiology reporting tasks while maintaining a fraction of the inference cost. Furthermore, EML operationalizes efficiency via cost-effective curriculum learning, as evidenced by LLaVA-Med (Li et al., 2023a),

which facilitates the rapid training of biomedical assistants in under 15 hours by transitioning from basic vocabulary alignment to complex reasoning. Beyond imaging, multimodal pathology models enable slide-level diagnosis, while speech- and sensor-based systems (Ji & Zhou, 2024; Ji et al., 2025a) support clinical monitoring. Ultimately, these efforts converge on a privacy-centric system orchestration, where system-level deployment on secure, on-site hardware necessitates the MAS synergy of lightweight fusion and verifiable reasoning to ensure scalable, interpretable, and ethically-compliant clinical intelligence.

### 6.5  Spatial Understanding

Spatial understanding, even physical understanding, represents a critical efficiency frontier where high-fidelity geometric perception—integrating LiDAR, camera, and radar signals—must operate within the rigid latency and energy envelopes of edge devices. The bottleneck in this domain lies in the inherent dimensionality of 3D sensor streams, necessitating a paradigm shift from dense volumetric fusion to projection-based efficiency. Frameworks like PointPillars (Lang et al., 2019) and BEVFusion (Liu et al., 2023b) operationalize this by projecting sparse 3D point clouds into compact 2D Bird's-Eye-View (BEV) grids, significantly reducing memory bandwidth and kernel launch overheads while preserving the geometric fidelity essential for safe navigation. This evolution highlights a deeper synergy between model topology and hardware-software co-design, where the architectural layout is natively aligned with the memory-access patterns of automotive-grade accelerators to prevent the "stalled pipeline" problem in multi-sensor synchronization. Furthermore, algorithmic innovations like FusionPainting (Xu et al., 2021) refine this process through sparse hybrid fusion, utilizing cross-modal semantic "painting" to minimize redundant computation by focusing resources on informative regions rather than exhaustive global processing. Together, these advancements move toward a heterogeneity-aware paradigm, where the system orchestrates the mapping of compute-dense modules to high-FLOP units and memory-bound operations to near-memory compute units, ultimately enabling real-time, efficient spatial intelligence that maintains high-resolution grounding under tight compute budgets.

### 6.6  Multimodal Reasoning

Multimodal reasoning marks the critical transition from sensory perception to symbolic intelligence, necessitating the synthesis of heterogeneous evidence for multi-step decision-making. The primary bottleneck of monolithic transformers in this domain is the propensity for semantic hallucinations and logic collapses during multi-step inference, which traditionally necessitates massive parameter redundancy to maintain stability. To address this, the field is shifting from dense statistical mapping—exemplified by early cross-attention models like UNITER (Chen et al., 2020)—toward modular functional decoupling. Frameworks such as ViperGPT (Surís et al., 2023) and VisProg (Gupta & Kembhavi, 2023) operationalize this by redefining reasoning as a programmatic execution problem, where the LLM serves as a high-level planner that invokes specialized, low-latency sub-routines only for necessary perceptual sub-goals. This structural evolution ensures that expensive generative resources are reserved for deliberative logic while routine sensory grounding is offloaded to efficient primitives, effectively embodying a "Thinking Fast and Slow" paradigm within the MAS system layer. Furthermore, in high-stakes domains such as mathematics and physics, specialized architectures like MathGLM-vision (Yang et al., 2024d) and ScienceQA (Lu et al., 2022) integrate verifiable process rewards to prune erroneous reasoning paths early, demonstrating how structured decomposition allows compact models to achieve high symbolic fidelity while minimizing the cumulative computational footprint of long-horizon tasks. Ultimately, these advancements move toward a neuro-symbolic synergy where efficiency is realized through the strategic orchestration of a hierarchical model stack, balancing expressive power with verifiable resource economy.

### 6.7  Federated Learning

Federated Learning (FL) provides a transformative framework for deploying multimodal intelligence within strictly regulated or sensitive environments such as medical diagnostics. While FL inherently addresses data sovereignty mandates by localizing raw data, achieving true protection against inference attacks requires it to be used in combination with differential privacy (Wei et al., 2020; Geyer et al., 2017). By training on local, non-independent and identically distributed data, such as hospital servers or mobile devices, this com-

bined approach reconciles the demand for sophisticated multimodal intelligence with the practical necessity of minimizing bandwidth costs and shielding user attributes from centralized exposure. To realize these applications, modern frameworks prioritize three functional pillars. Parameter-Efficient Federated Tuning methods bypass the prohibitive bandwidth of synchronizing massive backbones; systems like FedCLIP (Lu et al., 2023) and PFedPrompt (Guo et al., 2023) exchange only lightweight adapters or prompt vectors. This reduces communication payloads by orders of magnitude while retaining the generalization power of pre-trained models. Verifiable Privacy & Disentanglement methods realize the need for the integration of Differential Privacy (DP) and Modality-Specific Disentanglement. Approaches like FDARN (Yang et al., 2022) and MCARN (Yang et al., 2024c) separate modality-agnostic features from private, modality-specific noise to shield against gradient inversion attacks. A pivotal shift in recent benchmarks is the move toward Model-Heterogeneous Federated Learning. These methods accommodate missing-modality scenarios where clients have different sensor configurations. Frameworks like FedCMR (Zong et al., 2021) and PmcmFL (Bao et al., 2023) align local embeddings to shared prototypes, ensuring robustness even when cameras or microphones are unavailable on specific devices.

Current FL benchmarks are paving the way for a scalable, trust-less multimodal intelligence layer that operates efficiently under strict privacy and bandwidth budgets. FedMultimodal (Feng et al., 2023) provides a comprehensive testbed covering five distinct multimodal tasks under varying levels of data corruption and heterogeneity. Med-MMFL (Chhetri et al., 2026) evaluates multi-hospital collaborations across diverse modalities like MRI, ECG, and radiology reports. These methods and benchmarks establish a roadmap for moving multimodal learning out of the datacenter and into the secure, heterogeneous reality of edge-computing applications.

# 7 Holistic Discussion: Methodological Synthesis and Application-Driven Insights

The preceding analysis of model, algorithm, and system levels establishes that EML is no longer a collection of isolated optimization tricks, but a sophisticated exercise in cross-layer co-design. This section synthesizes these dimensions into a unified methodological framework, exploring how vertical synergy and application-specific constraints redefine the boundaries of multimodal intelligence.

## 7.1 The Mechanics of Vertical Synergy

The fundamental insight of the MAS framework lies in the realization that efficiency gains are non-linear; the most profound accelerations occur at the intersection of layers rather than within them. At the **model-algorithm interface**, we observe that structural decisions—such as the transition from dual-stream encoders to unified, sequence-based foundations—do not merely simplify the architecture but fundamentally reshape the optimization landscape for algorithmic compression. Unified foundations provide a homogeneous semantic space that mitigates the modality-specific outlier distributions that historically plagued quantization and pruning. Furthermore, the **algorithm-system nexus** reveals that theoretical compression (e.g., sub-4-bit quantization or token eviction) only translates into realized speedups when natively supported by hardware-aware execution kernels. The evolution from generic matrix multiplication toward specialized tensor-core utilization and kernel fusion highlights that algorithmic sparsity must be structured and hardware-aligned to bypass the memory-wall constraints of modern accelerators. Ultimately, true efficiency emerges when the model's structural topology is designed to be natively compatible with the system's execution dynamics, ensuring that every theoretical efficiency manifests as a measurable gain in wall-clock throughput.

## 7.2 Application-Driven Tactical Blueprints

The deployment of EML systems is rarely a pursuit of universal optimality; instead, it is a pragmatic navigation of the "Efficiency-Utility-Privacy" trilemma, dictated by the unique physical and regulatory constraints of specific domains. In **Latency-Critical environments** such as embodied AI and autonomous driving, the blueprint prioritizes structural-temporal symbiosis. Here, the system must utilize the edge-cloud continuum not as a simple storage tier, but as a hierarchical cognitive pipeline where low-level sensory filtering at the edge prevents bandwidth-induced decision stalls. Conversely, in **Fidelity-Critical domains** like medical imaging

and scientific discovery, the optimization focus shifts toward precision-preserving algorithms and federated system architectures. In these contexts, the methodological goal is to ensure that aggressive discretization does not erode the delicate semantic nuances required for diagnostic accuracy, while simultaneously utilizing decentralized learning to bypass the overhead of massive data aggregation. Finally, **Throughput-Oriented cloud services** demand a decoupling of capacity from cost, favoring structural sparsity (e.g., MoE) and aggressive cache management. By aligning the resource lifecycle with the momentary semantic complexity of user queries, these systems can sustain massive concurrent horizons within finite hardware budgets, demonstrating that the "optimal" MAS configuration is a dynamic, domain-specific equilibrium.

### 7.3 Reframing Efficiency: Toward Self-Regulating Intelligence

Looking beyond current methodologies, the synthesis of MAS layers points toward a fundamental reframing of efficiency: the transition from post-hoc adjustments to intrinsic, self-regulating properties. Historically, efficiency was treated as a secondary constraint—a "patch" applied to a pre-trained model for deployment. However, the emergence of natively efficient foundations and adaptive execution graphs suggests a future where intelligence and efficiency are inseparable. We envision a regime of cognitive-aware orchestration, where multimodal systems possess an internal representation of their own computational cost and hardware environment. In this paradigm, the model autonomously decides *what* information to process (Model), *how* to compress the computation (Algorithm), and *where* to execute the workload (System) based on real-time uncertainty and resource availability. This evolution toward self-regulating computation marks the final stage of the transition from models to systems, where efficiency is no longer an external metric to be minimized, but an emergent property of the model's fundamental structural design—an inherent parsimony that mirrors the biological efficiency of the human brain in processing the multimodal complexity of the physical world.

## 8 Open Challenges and Future Directions

Despite rapid progress in the EML domain, several challenges remain unresolved. We summarize key questions and outline promising directions to drive the next generation of scalable, deployable, and intelligent multimodal systems—across both understanding and generation tasks.

### 8.1 Unified Tokenization Across Modalities

Unified multimodal scaling faces a critical bottleneck in input representation: how heterogeneous signals such as images, audio, video, and text are converted into token-like units for shared processing. Importantly, this need not always be discrete or language-like. Many multimodal models already rely on continuous patch, frame, or latent embeddings, and future unified systems may reason directly over such representations without forcing all modalities through a text-centric interface. However, modality-specific tokenizers often produce semantically misaligned abstractions and highly uneven sequence lengths, which destabilize compute budgets and hinder efficient unified pretraining. Unified representation formation aligns diverse modalities in a shared semantic space while adaptively controlling granularity to satisfy latency and memory constraints. Promising directions include shared continuous latent spaces, shared discrete codebooks, variable-rate hierarchical compression, and tokenizer–scheduler co-design for KV-cache reuse and streaming inference.

### 8.2 Multimodal Multi-Task Generalization & Robustness

Current efficient models are often optimized for specific modality pairs (e.g., vision–language), leaving their transferability to new tasks or missing-modality scenarios unproven. A critical challenge is ensuring that efficiency gains are intrinsic to the architecture rather than overfitted to a specific dataset configuration. Future research must move beyond narrow, single-domain evaluations to establish cross-modal efficiency benchmarks. These should rigorously stress-test transferability—such as training on one modality set and evaluating resource-accuracy trade-offs on another—to ensure robust deployment across heterogeneous real-world conditions.

### 8.3 Hardware-Software Co-Design and Deployment

Bridging the gap between algorithmic efficiency and physical realization requires a paradigm shift toward hardware-software co-design. While optimization kernels for unimodal tasks (e.g., CNNs, LLMs) are mature, hardware support for multimodal interaction mechanisms—such as high-bandwidth cross-attention and dynamic modality switching—remains sparse. Future research must move beyond generic optimizations to develop multimodal-native accelerators and compiler-aware strategies, including operator fusion for fusion layers and hardware-friendly sparsity patterns. Furthermore, deploying on constrained platforms (mobile, AR/VR, robotics) necessitates robust edge–cloud orchestration to dynamically balance on-device latency with cloud-scale reasoning.

### 8.4 Human-Centric and Perceptual Efficiency

True efficiency extends beyond minimizing computational overhead to maximizing Quality of Experience (QoE) for the end-user. Current metrics (e.g., FLOPs, latency) often fail to capture human-centric constraints such as cognitive load, perceptual latency, and fairness. For instance, in real-time multimodal interaction, users are often sensitive to response fluidity and initial time-to-token rather than total throughput. A critical frontier lies in aligning system-level optimization with human perception thresholds. This involves redefining loss functions to penalize latency spikes that disrupt cognitive flow, and designing adaptive systems that trade off imperceptible fidelity drops for gains in interactivity. Future research must bridge the gap between system efficiency (resource usage) and user efficacy (satisfaction and interpretability).

### 8.5 Privacy-Aware Efficiency and Security

While efficient multimodal learning democratizes access to advanced capabilities, it introduces a double-edged sword: lowering the computational barrier paradoxically accelerates the potential proliferation of harmful applications, such as the scalable generation of deepfakes or automated misinformation. Furthermore, the pursuit of efficiency often necessitates architectural trade-offs that jeopardize security. For instance, aggressive compression can inadvertently widen the attack surface for model inversion.

Equally critical is the misconception regarding local deployment. While edge EML mitigates cloud data transmission risks, it is not automatically safe. Models localized on physical devices face severe vulnerabilities, including device-side model extraction, side-channel attacks, and physical tampering. Current research on jointly optimizing efficiency and privacy—such as secure aggregation (Bonawitz et al., 2017) or compressed encrypted inference (Mishra et al., 2020; Riazi et al., 2019)—remains sparse, particularly for high-dimensional, sensitive modalities like medical imaging and voice biometrics.

A critical future direction is the co-design of efficiency and privacy, moving beyond post-hoc defenses. This includes developing architectural sparsification (e.g., pruning, routing) that is intrinsically resistant to membership inference attacks, and designing multi-objective systems that dynamically optimize latency, energy, and privacy budgets against both cloud and edge threat models. Ultimately, the field must resolve the "Privacy-Efficiency-Utility" trilemma to enable trusted and responsible multimodal deployment.

### 8.6 Toward Standardized Benchmarks and Evaluation

A critical challenge hindering the progress of Efficient Multimodal Learning is the absence of standardized, deployment-aware benchmarking suites. While classic theoretical metrics—such as FLOPs, parameter counts, and theoretical MACs—remain indispensable for rapid algorithmic profiling and hardware-agnostic baseline comparisons, they are inherently insufficient for capturing the full spectrum of deployment realities. When used in isolation, these metrics often obscure memory-bound bottlenecks (e.g., KV cache I/O) and heterogeneous hardware overheads.

The fundamental difficulty in unifying evaluation stems from the extreme operational heterogeneity of multimodal applications and the rapid evolution of model architectures. As the field potentially evolves beyond the current LLM-centric paradigm toward non-autoregressive or continuous state-space models, static, language-centric evaluations will break down. A unified evaluation framework must gracefully balance diverse, and

sometimes conflicting, operational requirements. For instance, evaluating continuous streaming systems requires moving beyond discrete token generation to measure state-update latency and cross-modal synchronization overhead. Conversely, for real-time embodied agents, efficiency is defined by survivability, necessitating the measurement of end-to-end perception-action latency under strict thermal design power (TDP) constraints. Meanwhile, edge deployments require granular profiling of energy-per-sample (Joules/inference) and peak memory bandwidth utilization. Ultimately, the open challenge lies not in discarding classic metrics, but in integrating them into comprehensive, architecture-agnostic evaluation frameworks—akin to an "MLPerf for Multimodal Efficiency"—that evaluate models within simulated systemic pipelines rather than on isolated, static datasets.

## 9 Conclusion

This survey presents a comprehensive roadmap for Efficient Multimodal Learning, synthesizing over 300 studies into a unified Model–Algorithm–System taxonomy. We demonstrate that efficiency arises from the synergistic co-design of compact architectures, adaptive algorithms, and hardware-aware orchestration, rather than isolated optimizations. This reveals a critical paradigm shift: efficiency is evolving from a post-hoc constraint into an intrinsic design primitive. This work guides future work in navigating performance–cost trade-offs, paving the way for multimodal systems that are capable, robust, sustainable, and deployable across real-world applications.

## Broader Impact Statement

The rapid proliferation of multimodal learning, especially MLLMs, has ushered in a critical "efficiency wall", where the exponential demand for computational power and memory threatens the scalability, accessibility, and sustainability of multimodal intelligence. This survey, through the systematic lens of the MAS taxonomy, delivers a multi-faceted impact on the research community and broader society.

**1. Advancing the Academic Paradigm Toward Full-stack Co-design.** The primary intellectual merit of this work lies in transforming Efficient Multimodal Learning (EML) from a collection of isolated, modality-specific heuristics into a structured, engineering discipline. By establishing the MAS framework, this paper provides a unified blueprint for "vertical synergies"—encouraging researchers to move beyond single-layer optimizations toward hardware-aware architectural search and algorithm-system co-design. This holistic perspective is essential for demystifying the complexity of multimodal interactions and establishing a theoretical foundation for where and how efficiency can be injected without collapsing semantic fidelity.

**2. Democratization of High-Capability AI.** Multimodal intelligence is currently concentrated within resource-rich industrial laboratories due to the prohibitive costs of training and serving massive models. This survey fosters the democratization of AI by systematizing strategies like parameter-efficient fine-tuning (PEFT), knowledge distillation, and modular adaptation. These methodologies lower the barrier to entry, empowering researchers and developers with limited compute budgets to build and specialize high-performing multimodal systems. Furthermore, the emphasis on on-device EML ensures that advanced AI capabilities are no longer tethered to massive cloud infrastructures, allowing for pervasive, decentralized intelligence in various local environments.

**3. Environmental Sustainability and "Green AI".** As the carbon footprint of training and deploying large-scale AI becomes a global concern, the transition toward EML is a structural necessity for environmental sustainability. By promoting techniques that linearize quadratic complexity ($O(N^2) \rightarrow O(N)$) and maximize hardware utilization per watt, this work directly supports the global effort toward "Green AI". The focus on KV-cache reuse, token compression, and persistent state management reduces the cumulative energy consumption of long-horizon multimodal generation, making massive-scale deployment socially and environmentally responsible.

**4. Societal Safety, Privacy, and Human-Centric AI.** Beyond hardware metrics, the MAS framework emphasizes that computational efficiency must ultimately serve human-centric objectives. As EML systematically lowers deployment barriers, advanced multimodal models are increasingly migrating to ubiquitous edge

devices. In sensitive domains such as healthcare and affective computing, the paradigm shift toward on-device inference—enabled by algorithmic compressions like quantization and modular adaptation—fundamentally realizes "privacy-by-design" by retaining user data locally. However, this pervasive decentralization introduces an inherent security dichotomy: while it minimizes data-in-transit risks, it amplifies exposure to edge-specific vulnerabilities. Operating in physically accessible, low-power environments necessitates robust architectural safeguards to defend against adversarial model extraction and physical tampering, ensuring that ubiquitous intelligence strictly preserves user trust. Furthermore, in high-stakes arenas such as embodied AI and autonomous systems, computational efficiency is inextricably linked to physical safety. By drastically minimizing "perception-action latency", the optimization strategies outlined in this survey empower intelligent agents to process real-world stimuli in near-real-time, effectively mitigating the risk of catastrophic failures in highly dynamic and unpredictable environments.

In summary, this work offers a roadmap for the next generation of multimodal systems where efficiency is not a post-hoc optimization, but an intrinsic, self-regulating property that aligns artificial intelligence with the physical and ethical constraints of the human world.

## Acknowledgments

We thank the reviewers for their insightful comments, as well as Travis Allabon for his assistance with the initial literature collection. This work was partially supported by the National Science Foundation (NSF) under grant numbers CNS-2328972, CCF-2324864, CCF-2324937, CNS-2122320, and CNS-2133267, and by the National Institutes of Health (NIH) under grant numbers R01EB033387 and 1R01EB037101-01.

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

# Appendix

## A Case Study

To validate the descriptive and prescriptive power of the Model–Algorithm–System (MAS) framework, we conduct a retrospective analysis of a real-world edge multimodal system: the ultra-low-power audio–text emotion recognition pipeline by Mitsis et al. (Mitsis et al., 2025). While this system predates our framework, its design trajectory offers a compelling validation: under strict deployment constraints, the optimization choices naturally converge toward the principles codified in MAS. This case study demonstrates how the MAS framework serves as both an explanatory lens for existing successful designs and a principled blueprint for future resource-constrained multimodal systems.

### A.1 Model Level: Topology and Primitive Selection

**Hardware-Aligned Encoder Specialization.** The MAS model level dictates that architectural topology must be intrinsically decoupled from redundancy. Consistent with this, the case system avoids generic large-scale backbones, opting for *modality-specific specialization*: a compact transformer with a CNN front-end for acoustics, and a lightweight keyword-spotting encoder for text. These choices reflect core MAS design primitives: early dimensionality reduction, the use of depthwise separable convolutions, and unified embedding dimensions to facilitate low-cost fusion.

**Minimalist Late Fusion.** To balance expressivity with latency, the system employs a fusion strategy that strictly separates representation learning from cross-modal reasoning—a key MAS recommendation for edge devices. The fusion head is implemented as a lightweight two-layer MLP:

$$h_{\text{cat}} = \text{Concat}(h^{(a)}, h^{(t)}), \quad \hat{y} = \text{Softmax}(W_2, \sigma(W_1 h_{\text{cat}})), \tag{6}$$

where $\sigma$ denotes the activation function. This design minimizes parameters and avoids the quadratic complexity of cross-attention, ensuring the fusion mechanism remains agnostic to input sequence length and friendly to cache-limited hardware.

**Constraint-Aware Primitive Selection.** The MAS framework emphasizes that "efficiency starts at design time". This is evident in the system's architectural constraints: the exclusive use of ReLU6 activations, single-head attention mechanisms, and static tensor shapes. These are not merely algorithmic preferences but model-level structural decisions explicitly chosen to map efficiently to the specific instruction set architecture (ISA) of the target Edge TPU.

### A.2 Algorithm Level: Flow Modulation and Robustness

**Pre-Deployment Adaptation.** The training pipeline exemplifies the MAS principle of "optimizing information flow before physical mapping". Rather than relying solely on post-training compilation, this work integrates algorithmic constraints—such as quantization-aware training (QAT)—directly into the learning loop. This ensures that the model's weights learn to accommodate the precision loss inherent to 8-bit integers.

**Algorithmic Substitution for Model Capacity.** Instead of scaling up model depth to handle noisy real-world data, the system leverages algorithm-level robustness strategies. Aggressive domain-targeted augmentations (e.g., frequency masking, shifting) and label smoothing are employed to mitigate the domain shift between PC-recorded training data and MCU-recorded inference data. From an MAS perspective, this demonstrates how *algorithmic complexity* (during training) can effectively substitute for *model capacity* (during inference), maintaining high accuracy without inflating the parameter budget.

### A.3 System Level: Execution Orchestration

**Train–Inference Consistency.** A recurring failure mode in multimodal deployment is the misalignment between offline data loaders and online feature extractors. The case system adheres to the MAS requirement

for *pipeline coherence* by utilizing the Silicon Labs MLTK MicroFrontend for both training and on-device inference. This ensures bit-exact consistency in spectrogram generation, eliminating a common source of system-level degradation.

**Hardware-Software Co-Design.** The system demonstrates the tight coupling between model architecture and hardware constraints characterized by MAS. The Edge TPU imposes strict limits: INT8-only execution, unbatched 3D tensors, and specific supported operations. The design process reflects a bidirectional optimization loop: system constraints dictated the removal of complex attention heads, while model requirements drove the selection of specific compiler directives.

**Quantifiable Efficiency Gains.** The final deployed system satisfies multidimensional resource budgets, validating the effectiveness of the MAS-aligned approach:

- **Latency:** ≈22 ms per inference (Real-time),

- **Storage:** 1.8 MB (INT8 quantized),

- **Memory footprint:** ≈1.4 MB peak RAM,

- **Energy:** ≈2.5 W active power.

### A.4 Conclusion: MAS as a Blueprint

This analysis confirms that high-efficiency multimodal systems are not products of isolated optimizations but of vertical synergy. The case study illustrates the hierarchical reasoning inherent to MAS, which naturally yields a deployable multimodal system under real hardware constraints.

- **Model:** Define *what* to compute by selecting hardware-friendly, modality-specific topologies.

- **Algorithm:** Determine *how* to compute by embedding quantization and robustness constraints into the learning objective.

- **System:** Decide *where* to compute by enforcing pipeline consistency and respecting accelerator-specific operation sets.

## B Subset of Recent Efficient Multimodal Models

Tables 3 and 4 present a curated subset of recent efficient multimodal models organized under the MAS framework. Our goal is not to enumerate all existing systems, but to highlight representative architectures that capture the dominant design patterns in practice.

## C Comparison with Existing Surveys

To clearly position the novelty and comprehensiveness of this work, we compare our survey with recent related reviews in the fields of Multimodal Large Language Models (MLLMs), Vision-Language Models (VLMs), and resource-efficient AI. As summarized in Table 5, while existing surveys provide excellent coverage of specific mechanisms (e.g., token compression, edge deployment) or task-oriented applications, they typically adopt a fragmented approach.

Our work is the first to introduce a vertical, full-stack **Model-Algorithm-System (MAS)** taxonomy. Notably, our survey bridges the critical gap between high-level architectural designs and underlying physical execution constraints (e.g., KV cache management, hardware-aware scheduling, and edge-cloud orchestration), offering a unified blueprint for Efficient Multimodal Learning (EML) that is absent in prior literature.

Table 3: Representative efficient multimodal models summarized under the MAS framework in recent years. V: Video/Image, T: Text, A: Audio, MSM: multiple sensor modalities, Act: Action, Clin: Clinical, TS: Time series, Geo: location coordinates.

| Model | Backbone | Year | Parameters | Efficient strategy | Modalities |
|---|---|---|---|---|---|
| Cobra (Zhao et al., 2025a) | Mamba, DINOv2-L, SigLIP | 2025 | 2.8B/7B | Modality Specific | V+T |
| LLaVA-Gemma (Hinck et al., 2024) | CLIP ViT-L+Gemma | 2024 | 2B/7B | Modality Specific | V+T |
| FW-SAT (Jiang & Chen, 2024) | Swin-Transformer–style | 2024 | - | Modality Specific | V+Thermal |
| UGen (Tang et al., 2025) | TinyLlama | 2025 | 1.1B | Unified | V+T |
| i-Code V2 (Yang et al., 2024f) | OmniVL, WavLM Large, Z-Code | 2024 | 1.4B | Unified | V+T+Speech |
| Unified-IO 2 (Lu et al., 2024a) | Transformer+ViT-based | 2024 | 1.1B/3.2B/6.8B | Unified | V+T+A+Act |
| UNIT (Zhu et al., 2024b) | ViT-H | 2024 | 632M | Unified | V+T |
| Emu (Sun et al., 2024) | EVA-01-CLIP + LLaMA | 2024 | 14B | Unified | V+T |
| Grok-1.5V (xAI / Grok team, 2024) | Grok-1.5 LLM | 2024 | - | Unified | V+T |
| Astrea (Yang et al., 2025e) | Vicuna-1.5, Hermes2-Yi | 2025 | 13B/34B | Structural Sparsity | V+T |
| FMT (Xu & Wang, 2025) | ResNet-50, BERT | 2025 | - | Structural Sparsity | V+T |
| LEO-MINI (Wang et al., 2025f) | Llama3 | 2025 | 8B | Structural Sparsity | V+T |
| NVILA (Liu et al., 2025b) | SigLIP, Qwen2 | 2025 | 8B/15B | Structural Sparsity | V+T |
| SmolVLM (Marafioti et al., 2025) | SigLIP, SmolLM2 | 2025 | up to 2.2B | Structural Sparsity | V+T |
| Elastic EVE (Rang et al., 2025) | PanGu, ResNet-50, SigLIP, ViT-L | 2025 | 1.8B | Structural Sparsity | V+T |
| MoMa (Lin et al., 2024d) | Early-fusion multimodal Transformer | 2024 | 1.4B/2.3B | Structural Sparsity | V+T |
| MoME (Shen et al., 2024) | Vicuna-7B, CLIP, DINO, Pix2Struct | 2024 | 7B | Structural Sparsity | V+T |
| LLaVA-MoLE (Chen et al., 2024d) | LLaVA-1.5 | 2024 | 7B | Structural Sparsity | V+T |
| MoE-LLaVA (Lin et al., 2024a) | MoE-LLaVA | 2024 | 1.6B/1.8B/2.7B | Structural Sparsity | V+T |
| EVE (Chen et al., 2024c) | BEiTv2-initialized | 2024 | - | Structural Sparsity | V+T |
| RoE (Wu et al., 2025c) | LLaVA-1.5, LLaVA-HR, VILA | 2024 | 7B | Structural Sparsity | V+T |
| CuMo (Li et al., 2024a) | Mistral/Mixtral-8, CLIP-L | 2024 | 7B–13B | Structural Sparsity | V+T |
| Flex-MoE (Yun et al., 2024) | Transformer | 2024 | 37M | Structural Sparsity | Clin |
| Fuse-MoE (Han et al., 2024) | Longformer, Transformer, CNN, DenseNet | 2024 | - | Structural Sparsity | V+Clin+TS |
| Omni-SMoLA (Wu et al., 2024a) | PaLI-X/PaLI-3 | 2024 | 5B/55B | Structural Sparsity | V+T |
| Wander (Guo et al., 2025b) | BERT-base+ViT | 2025 | 80-220M | Adapters | V+T+A |
| SALM (Chen et al., 2024e) | Fast Conformer,Megatron LLM, modality adapter | 2024 | 2B | Adapters | Speech+A+T |
| Enhancing-LoRA (Ji et al., 2025b) | BLIP | 2025 | 223M | Adapters | V+T |
| CROME (Ebrahimi et al., 2024) | ViT-G, Vicuna, Flan-T5-XXL | 2024 | 7B/13B/11B | Adapters | V+T |
| MMA (Yang et al., 2024b) | CLIP | 2024 | - | Adapters | V+T |
| PaLM2-VAdapter (Xiao et al., 2024) | CoCa ViT, PaLM 2 | 2024 | 1.8B/2.0B/2.8B/10.8B | Adapters | V+T |
| ST3 (Zhuang et al., 2025) | LLaVA-1.5 | 2025 | 7B/13B | Token Compression | V+T |
| DART (Wen et al., 2025) | LLaVA-1.5/NEXT, Video-LLaVA, Qwen2-VL, MiniCPM | 2025 | 7B/8B | Token Compression | V+T |
| TopV (Yang et al., 2025b) | LLaVA-1.5, Inern-VL2, Video-LLaVA | 2025 | 2B/7B/13B/26B | Token Compression | V+T |
| VisionZip (Yang et al., 2025d) | LLaVA-1.5, LLaVA-NEXT | 2025 | 7B/13B | Token Compression | V+T |
| TokenCarve (Tan et al., 2025) | LLaVA-1.5 | 2025 | 7B/13B | Token Compression | V+T |
| AdaRETAKE (Wang et al., 2025e) | LLaVA-Video, Qwen2/2.5-VL | 2025 | 7B/72B | Token Compression | V+T |
| Audio token compression LALM (Bhati et al., 2025) | Qwen2-Audio, token compression module, LoRA | 2025 | 7B | Token Compression | A+T |
| OmniZip (Tao et al., 2025) | Qwen2.5-Omni | 2025 | - | Token Compression | A+V+T |
| HiRED (Arif et al., 2025) | ShareGPT4V, LLaVA-Next, LLaVA-1.5 | 2025 | 7B/13B | Token Compression | V+T |
| Fit-and-Prune (Ye et al., 2025b) | LLaVA-1.5, LLaVA-NEXT, LLaVA-HR | 2025 | 7B | Token Compression | V+T |
| Qwen-Audio (Chu et al., 2023) | Whisper-large-v2, Qwen LLM | 2023 | 7.7B | Token Compression | A+T |
| ViLA (Wang et al., 2024d) | ViT-G, Flan-T5 XL | 2024 | 4B | Token Compression | V+T |
| TOKEN (Omri et al., 2025) | LLaVA-1.5, VILA | 2025 | 7B/13B/8B | Token Compression | V+T |
| Folder (Wang et al., 2025b) | LLaVA-1.5, MiniGPT4v2, MMVP, Video-LLaVA, BLIP | 2025 | 7B/13B | Token Compression | V+T |
| FAST (Pertsch et al., 2025) | π0 VLA, OpenVLA | 2025 | 3B/7B | Token Compression | V+T+Act |
| ZipVL (He et al., 2024b) | LLaVA-Next/1.5, LongVA, Qwen-VL | 2024 | 7B/13B | Token Compression | V+T |
| MUST-Drop (Liu et al., 2024a) | LLaVA-1.5, LLaVA-Next, Video-LLaVA | 2024 | 7B | Token Compression | V+T |
| LLaVolta (Chen et al., 2024a) | CLIP ViT-L/14, Vicuna-v1.5 | 2024 | 7B | Token Compression | V+T |
| PyramidDrop (Xing et al., 2024) | LLaVA-NeXT, LLaVA-1.5 | 2024 | 7B | Token Compression | V+T |
| GPrune (Jiang et al., 2025) | LLaVA-NeXT | 2024 | 8B | Token Compression | V+T |
| P-Mod (Zhang et al., 2025a) | LLaVA-1.5, LLaVA-NeXT | 2024 | 7B | Token Compression | V+T |
| TokenPacker (Li et al., 2025d) | LLaVA-1.5 | 2025 | 7B/13B | Token Compression | V+T |
| DeepStack (Meng et al., 2024) | Vicuna, CLIP | 2024 | 7B/13B | Token Compression | V+T |
| MoPE-CLIP (Lin et al., 2024b) | CLIP-ViT-B/32, SE-CLIP | 2024 | 194M | Pruning | V+T |
| MANU (Liu et al., 2025a) | LLaVA-1.5, Idefics2 | 2025 | 7B/8B | Pruning | V+T |
| EfficientLLaVA (Liang et al., 2025a) | LLaVA-v1.5/LLaVA-SQA | 2025 | 7B+ | Pruning | V+T |
| Q-VLM (Wang et al., 2024b) | MoE-LLaVA, LLaVA | 2025 | 7B/13B/1.6B | Quantization | V+T |
| MBQ (Li et al., 2025c) | LLaVA-onevision, InternVL2, Qwen2-VL | 2025 | 7B/8B/26B/72B | Quantization | V+T |
| MQuant (Yu et al., 2025) | InternVL2, Qwen/2-VL, MiniCPM, GLM, | 2025 | 8B/9.6B/7B/9B/72B | Quantization | V+T |
| VLMQ (Xue et al., 2025) | Qwen2-VL, Qwen2.5-VL, LLaVA-onevision | 2025 | 7B/2B | Quantization | V+T |
| STaMP (Federici et al., 2025) | Qwen 2.5, Llama3/3.2, PixArt-Σ, SANA | 2025 | 0.6B/1.6B/1B/3B/8B | Quantization | V+T |
| SpeechTokenizer (Zhang et al., 2023c) | EnCodec-based RVQ-GAN codec + 2-layer BiLSTM | 2024 | - | Quantization | Speech + A |
| Retrieval-based Biasing (Flemotomos et al., 2025) | CTC-AED | 2025 | - | Quantization | Speech+A+T |
| QSLAW (Xie et al., 2024) | CLIP-ViT-L, LLaMA, Vicuna | 2024 | 7B–13B | Quantization | V+T |

Table 4: Representative efficient multimodal models summarized under the MAS framework in recent years. V: Video/Image, T: Text, A: Audio, MSM: multiple sensor modalities, Act: Action, Clin: Clinical, TS: Time series, Geo: location coordinates.

| | Model | Backbone | Year | Parameters | Efficient strategy | Modalities |
|---|---|---|---|---|---|---|
| | DHO (Kang et al., 2025) | CLIP ViT-B/L, ResNet, MobileNetV2, DFN ViT-H | 2025 | 3.5M-304M | Knowledge Distillation | V+T |
| | FDBPL (Zhang et al., 2025c) | CLIP-style, ViT-L/14, ViT-B/32 | 2025 | - | Knowledge Distillation | V+T |
| | Comodo (Chen et al., 2025a) | TimeSformer, Mantis, MOMENT | 2025 | 150M | Knowledge Distillation | V+Motion |
| | MoveKD (Cao et al., 2025) | LLaVA-1.5, LLaVA-NeXT | 2025 | 1.7B/7B/13B | Knowledge Distillation | V+T |
| | mm-Mamba (Liao et al., 2025) | mmMamba-linear, mmMamba-hybrid | 2025 | 2.7B | Knowledge Distillation | V+T |
| | OpenVoca (Wei et al., 2025) | ResNet-50, ResNet-152, EfficientNet | 2025 | - | Knowledge Distillation | V+T |
| | CLIP-CID (Yang et al., 2025c) | ViT-B/32, ViT-B/16, OPENCLIP ViT-bigG/1 | 2025 | - | Knowledge Distillation | V+T |
| | Skywork R1V (Peng et al., 2025) | Skywork R1 | 2025 | 38B | Knowledge Distillation | V+T |
| | TAID (Shing et al., 2025) | Qwen2, InternVL2 | 2025 | 8B/ 72B | Knowledge Distillation | V+T |
| | LiSER (Pendyala et al., 2025) | 2D CNN, LSTM | 2025 | 105K | Knowledge Distillation | Speech+A |
| | PromptKD (Li et al., 2024e) | ViT-B/16, ViT-L/14, CLIP | 2024 | - | Knowledge Distillation | V+T |
| Algorithm Level | DSMD (Liang et al., 2025b) | VIT, BERT | 2024 | 197M | Knowledge Distillation | V+T |
| | AMFD (Chen et al., 2025d) | ResNet-18 | 2024 | - | Knowledge Distillation | V+MSM |
| | CLIP-KD (Yang et al., 2024a) | ViT-B, CLIP-ViT-L | 2024 | 350M | Knowledge Distillation | V+T |
| | LLavaKD (Cai et al., 2025) | SigLIP-B, Qwen1.5 | 2024 | 7B | Knowledge Distillation | V+T |
| | LLaVA-MoD (Shu et al., 2024) | CLIP ViT-L/14, Qwen-1.5/2 | 2024 | 8B | Knowledge Distillation | V+T |
| | VPD (Hu et al., 2024) | PaLI-3/PaLI-X | 2024 | 5B/55B | Knowledge Distillation | V+T |
| | MEDA (Wan et al., 2025) | LLaVA-family, InternVL, LongVA | 2025 | 7B–32B | Caching & Reuse | V+T |
| | FastCache (Zhu et al., 2025) | LLaVA-1.5 | 2025 | 7B | Caching & Reuse | V+T |
| | AirCache (Huang et al., 2025) | LLaVA-OneVision, InternVL2, Qwen2-VL | 2025 | 1B/4B/7B/8B/26B | Caching & Reuse | V+T |
| | VLA-Cache (Xu et al., 2025) | OpenVLA, OpenVLA-OFT, CogAct | 2025 | - | Caching & Reuse | V+T+Act |
| | VL-Cache (Tu et al., 2024) | llava-v1.6-mistral, llava-v1.6 | 2024 | 7B/34B | Caching & Reuse | V+T |
| | PrefixKV (Wang et al., 2024a) | LLaVA-1.5 | 2024 | 7B/13B | Caching & Reuse | V+T |
| | ElasticCache (Liu et al., 2024c) | LLaVA-1.5, Qwen2-VL | 2024 | 7B/13B | Caching & Reuse | V+T |
| | CSP (Pei et al., 2024) | InceptionV3, ResNet-50, Space2Vec | 2024 | 40M | Caching & Reuse | V+Geo |
| | LOOK-M (Wan et al., 2024) | LLaVA-v1.5, InternVL-v1.5, MobileVLM-v2 | 2024 | 3B/7B/13B | Caching & Reuse | V+Chat |
| | AASD (Yang et al., 2025a) | LLaVA-1.5, LLaVA-NeXT, InternVL-1.5 | 2024 | 7B/13B | Speculative Decoding | V+T |
| | SpecVLA (Wang et al., 2025d) | OpenVLA | 2025 | 7B | Speculative Decoding | V+T+ACT |
| | λ-MoD (Luo et al., 2024) | LLaVA-1.5, LLaVA-HR, Mini-Gemini-HD | 2024 | 7B/13B | Runtime Sparsity | V+T |
| | LoRA-Sparse (Song et al., 2024) | LLaVA-1.5 | 2024 | 7B/13B | Runtime Sparsity | V+T |
| | LLaVA-PruneMerge (Shang et al., 2025) | LLaVA-1.5, Video-LLaVA | 2025 | 7B/13B | Runtime Sparsity | V+T |
| | SkipVision (Zeng et al., 2025) | LLaVA, LLaVA-HD, CoS | 2025 | 8B | Runtime Sparsity | V+T |
| | HuBERT-EE (Yoon et al., 2024) | HuBERT | 2024 | 95M | Runtime Sparsity | Speech + A |
| | MoLe-VL (Zhang et al., 2025b) | OpenVLA, CogAct VLA | 2025 | 7B | Runtime Sparsity | V+T+Act |
| | DeeVISum (Khan et al., 2025) | PaLI-Gemma2 VLMs | 2025 | 3B/10B/28B | Runtime Sparsity | V+T+A |
| | MEDA (Wan et al., 2025) | LLaVA-family, InternVL, LongVA | 2025 | 7B–32B | Cache Management | V+T |
| | FastCache (Zhu et al., 2025) | LLaVA-1.5 | 2025 | 7B | Cache Management | V+T |
| | MPIC (Zhao et al., 2025c) | LLaVA-1.6 | 2025 | 7B | Cache Management | V+T |
| | Cache-of-Thought (Wu et al., 2025a) | GPT-4o, Qwen-VL-2, OpenFlamingo | 2025 | 3B/7B/9B | Cache Management | V+T |
| | CEAM-GM (Lee et al., 2025b) | Code Llam, SeamlessM4T, Chameleon | 2025 | 34B | Cache Management | V+T+Speech |
| | video-SALMONN S (Sun et al., 2025a) | Qwen3-VL, Whisper-Large-v3, window-level Q-Former | 2026 | 8B | Cache Management | A+V+T |
| System Level | Gemini 1.5 (Team et al., 2024) | Gemini 1.5 Pro/Flash | 2024 | - | Cache Management | V+T+A |
| | CloudEdgeCo (Wang et al., 2025a) | any ReID, CNN | 2025 | - | Edge–cloud Collaboration | V+TS |
| | MoA (Yang et al., 2025f) | Qwen2-VL-2B, Qwen2.5-VL | 2025 | 2B/7B | Edge–cloud Collaboration | V+T |
| | EdgeCloudAI (Ghasemi et al., 2024) | cloud VLM, CNN | 2024 | - | Edge–cloud Collaboration | V+T |
| | S2ST (Zhu et al., 2026) | Whisper-Large-V, Flan-T5 SLM, IndexTTS2 | 2026 | - | Edge–cloud Collaboration | Speech+A |
| | RServer (Guo et al., 2025a) | Qwen2.5-VL | 2025 | 72B | Job Scheduling | V+T |
| | Amazon Nova (Langford et al., 2025) | Nova | 2025 | - | Job Scheduling | V+T |
| | ProxyV (Wu et al., 2025b) | Vicuna-1.5 InternLM2.5 | 2025 | 7B | Job Scheduling | V+T |
| | DivPrune (Alvar et al., 2025) | LLaVA-1.5, LLaVA-1.6 | 2025 | 7B | Job Scheduling | V+T |
| | Inf-MLLM (Ning et al., 2024) | Vicuna, LLaMA-2, Pythia, Chat-UniVi, Flash-VStream | 2024 | 7B | Job Scheduling | V+T |

Table 5: Comparison of our survey with existing related reviews. **M**: Model architecture optimization. **A**: Algorithm-level compression/modulation. **S**: System-level orchestration and hardware deployment. (✓: Comprehensive coverage; ○: Partial or brief mention; ×: Not covered)

| Survey | Primary Focus | M | A | S | Taxonomy Basis |
|---|---|---|---|---|---|
| Efficient LLM (Zhou et al., 2024) | Efficient LLMs (Text-only) | ✓ | ✓ | ✓ | Data-Model-System pipeline |
| Advances in MLLMs (Zhang et al., 2024a) | MLLM Architectures & Tasks | ✓ | × | × | Task formulations & Capabilities |
| Token Compression (Yao et al., 2025) | Token Compression for MLLMs | × | ✓ | × | Component location (Encoder/LLM) |
| Edge VLM (Sharshar et al., 2025) | VLMs for Edge Networks | ○ | ✓ | ✓ | Device constraints (Mobile, IoT) |
| **Ours (MAS)** | **Full-stack Efficient Multimodal Learning** | ✓ | ✓ | ✓ | **Model-Algorithm-System** |

