# OpenReview forum: "From Models to Systems: A Comprehensive Survey of Efficient Multimodal Learning"
_TMLR — Accepted by TMLR_

### Review · Reviewer_vrze · 2026-02-22

**Summary Of Contributions:**

This paper presents a highly comprehensive and timely survey on Efficient Multimodal Learning (EML) by introducing a structured Model-Algorithm-System (MAS) taxonomy. The authors synthesize insights from over 300 recent works, offering a valuable end-to-end perspective on how multimodal models are optimized for computation, memory, and deployment. The primary strength of this submission lies in its extensive literature collection and its practical engineering focus, particularly the detailed discussions on KV cache management and edge-cloud orchestration. While the survey excels in mapping the system-level and algorithmic landscape, the theoretical discussions regarding the fundamental limits of multimodal compression could be further enriched to match the depth of the engineering sections. Overall, it serves as an excellent resource for researchers and practitioners navigating the complex EML ecosystem.

**Additional Comments:**

This is a remarkably broad and well-organized survey that provides a solid foundation for understanding the EML landscape. The MAS taxonomy is intuitive and helpful. The suggested changes are primarily aimed at adding a touch of theoretical nuance to complement the excellent systemic overview. I recommend leaning towards acceptance, as this work will undoubtedly be a valuable reference for the TMLR audience.

**Audience:**

Yes

**Audience Explanation:**

The memory wall and deployment bottlenecks of large multimodal models are pressing challenges in the community. Researchers engineering foundational models, deploying autonomous agent systems, or working on resource-constrained edge intelligence will find the structured insights on asynchronous pipelining, structural sparsity, and hardware-software co-design highly practical and inspiring.

**Broader Impact Concerns:**

The submission includes a thoughtful Broader Impact Statement that adequately addresses environmental sustainability and the democratization of AI. To make this section even more robust, the authors could briefly expand on the security implications of edge deployment. As EML successfully lowers the computational barrier, making it easier to deploy advanced vision-language models on everyday devices, discussing the importance of privacy safeguards in such ubiquitous, low-power environments would provide a very well-rounded ethical perspective.

**Claims And Evidence:**

Yes

**Claims Explanation:**

The core claims regarding the operational mechanics of various EML techniques are factually accurate and well-grounded in the extensively cited literature. The MAS framework effectively categorizes the current state of the field. However, the assertion in the abstract that the cross-layer co-design framework "resolves" the efficiency-utility-privacy trilemma might be slightly strong. Given that many current optimization techniques rely on empirical trade-offs and heuristic scheduling under specific hardware constraints, it would be more precise to state that the framework helps to "navigate" or "mitigate" these trade-offs rather than providing a definitive resolution.

**Requested Changes:**

To further strengthen this already impressive survey, a few minor adjustments are recommended. First, it would be beneficial to slightly soften the terminology around the efficiency-utility-privacy trilemma, perhaps framing it as a balance rather than a solved problem. Second, while the discussion on structural sparsity and modular adaptation is thorough, the paper would benefit from a brief theoretical discussion on the representational trade-offs of these methods. For instance, gently touching upon how aggressive compression or parameter-efficient fine-tuning might impact the model's original feature space or generalization capabilities would add valuable depth. Finally, to make the comparative tables even more informative, the authors might consider incorporating formal complexity bounds (such as contrasting $O(N)$ and $O(N^2)$ theoretical limits) alongside the empirical parameter counts, which would better highlight the mathematical efficiency gains of different architectures

---

> ### Author Response · Authors · 2026-03-22
> **Responses to Comments from Reviewer vrze**
>
> Dear Reviewer vrze,
>
> We sincerely thank you for your positive evaluation and constructive feedback. We have carefully revised the manuscript based on your suggestions. Please find our responses below (all revisions are highlighted in blue in the revised manuscript):
>
> **[1] Softening the terminology around the efficiency-utility-privacy trilemma**
>
> We have revised the abstract and introduction to frame the trilemma as an empirical balance rather than a solved problem. Specifically, we replaced definitive terms (like "resolves") with more nuanced language such as **"contributes to"** and **"mitigates"** to accurately reflect that the framework helps navigate these trade-offs.
>
> Revised Text Excerpt (Abstract): "...elucidating how cross-layer co-design contributes to the fundamental 'Efficiency-Utility-Privacy' trade-off."
>
> Revised Text Excerpt (Introduction / MLLM section): "...demonstrating how the convergence of perception, execution, and scheduling mitigates the fundamental 'Efficiency-Utility-Privacy' trilemma."
>
> **[2] Theoretical discussion on representational trade-offs**
>
> We have integrated theoretical discussions directly into the respective subsections to explicitly address the impact on feature space and generalization:
>
> **Structural Sparsity**: We discussed how discrete routing causes representational fragmentation.
>
> Revised Text Excerpt: "...this mathematical decoupling of parameters from FLOPs inherently risks representational fragmentation... Consequently, while in-domain performance scales efficiently, zero-shot generalization may degrade when confronted with complex multimodal inputs that lack a dense, unified pathway..."
>
> **Structural Decoding**: We discussed how compressing visual sequences creates an information bottleneck.
>
> Revised Text Excerpt: "...This many-to-few mapping behaves analogously to a low-pass filter on the feature space, smoothing out high-frequency spatial details. Therefore... they often struggle with tasks requiring dense, pixel-level grounding..."
>
> **Modular Adaptation**: We discussed how extreme low-rank constraints mathematically limit capacity.
>
> Revised Text Excerpt: "...this extreme low-rank constraint limits the model's capacity to internalize entirely new, orthogonal multimodal knowledge... occasionally leading to degraded generalization on out-of-distribution (OOD) domains."
>
> **[3] Incorporating formal complexity bounds in comparative tables**
>
> We have thoroughly updated Table 1 by adding explicit "Time Complexity" and "Space Complexity" columns. The table now formally contrasts the $\mathcal{O}(\cdot)$ theoretical limits (e.g., $\mathcal{O}((N+M)^2 \cdot D)$ for the baseline vs. $\mathcal{O}((r \cdot N + M)^2 \cdot D)$ for token reduction) alongside the empirical parameter counts for all discussed mechanisms. All these formal comparisons are now directly discussed in the corresponding theoretical text.
>
> **[4] Expanding on the security implications of edge deployment**
>
> We have expanded Point 4 in the section of Broader Impact Statement to address the "security dichotomy" of edge deployment, emphasizing the critical need for robust architectural safeguards.
>
> Revised Text Excerpt (Point 4 in the section of Broader Impact Statement):
>
> "...this pervasive decentralization introduces an inherent security dichotomy: while it minimizes data-in-transit risks, it amplifies exposure to edge-specific vulnerabilities. Operating in physically accessible, low-power environments necessitates robust architectural safeguards to defend against adversarial model extraction and physical tampering..."
>
> Thanks again. We hope these revisions fully address your valuable feedback.
>
> Best regards,
>
> The Authors

---

### Review · Reviewer_nC85 · 2026-03-23

**Summary Of Contributions:**

This paper presents a broad survey of Efficient Multimodal Learning (EML) and organizes the space through a Model-Algorithm-System (MAS) taxonomy. The central idea is to move beyond fragmented, mechanism-level discussions of efficiency and instead ask where efficiency enters the multimodal stack: model-level design choices that define what to compute, algorithm-level methods that define how to compute, and system-level orchestration that defines where and when to compute. The survey also includes an extended discussion of efficient MLLMs, application-specific perspectives, open challenges, and a broader-impact discussion. The paper explicitly positions its main contribution as a unified taxonomy of over 300 works, a cross-level analysis of interactions between layers, an MLLM-focused synthesis, and a roadmap for future research.

I think the paper’s main strength is that it tries to elevate efficiency in multimodal learning from a bag of tricks into a full-stack perspective. The MAS framing is intuitive and easy to follow, and Figures 1-2 give a useful visual map of the space. The paper is also broad in scope: it covers model design, pruning/quantization/token compression/caching, deployment issues such as scheduling and edge-cloud execution, and then ties these to applications like healthcare, robotics, affective computing, and media understanding. The discussion of trade-offs at several points, such as structural bottlenecks, routing fragmentation, and low-rank adaptation limits, is also positive because the paper is not purely celebratory. The taxonomy and accompanying representative-work tables would likely be useful to newcomers and to practitioners trying to place a method in the broader landscape.

The main weakness is that the paper makes some fairly strong positioning claims that are not fully backed by a systematic survey methodology or by sufficiently careful comparative evidence. For example, the paper repeatedly claims to provide the “first” structured model-to-system taxonomy and frames MAS as a formal foundation for the area, but I did not see a rigorous comparison against prior surveys that would make this novelty claim airtight. Similarly, some of the higher-level claims about mitigating the “Efficiency-Utility-Privacy” trilemma or about a paradigm shift toward “self-regulating intelligence” read more like a forward-looking perspective than conclusions established by evidence. The survey is also sometimes listing-heavy: many sections enumerate representative methods but do not consistently compare them under unified axes such as training cost, inference latency, memory footprint, accuracy retention, deployment regime, or modality setting. As a result, the paper is strongest as a conceptual map, but weaker as a systematic empirical synthesis.

**Additional Comments:**

Overall, I think this is a potentially useful survey with a strong organizing idea. The MAS framing is good, the topic is important, and I can see the paper being useful to the community. My reservation is mainly about the rigor of support for the strongest claims. If the authors strengthen the survey methodology, clarify novelty relative to prior surveys, and make the synthesis more systematically comparative rather than mostly narrative, the paper would become substantially stronger.

If you want, I can also turn this into a more polished final-review version with a likely score/confidence and a more acceptance-oriented or rejection-oriented tone.

**Audience:**

Yes

**Audience Explanation:**

Yes. Efficient multimodal learning is a timely and important topic, and there is real value in a survey that tries to unify architecture, compression/acceleration, and systems concerns rather than treating them separately. TMLR’s audience includes multimodal learning researchers, systems researchers, and practitioners developing deployable MLLMs, all of whom could benefit from a full-stack taxonomy such as MAS. The paper also addresses several applications in which efficiency is essential, including healthcare, robotics, streaming media, and embodied AI, thereby strengthening its relevance.

I also think the paper would be useful to readers because it does more than catalogue model-level tricks. It explicitly covers KV cache management, edge-cloud collaboration, latency-aware scheduling, federated learning, and hardware-software co-design. That broader systems emphasis is likely to interest a meaningful subset of the TMLR readership, especially as the community increasingly cares about deployment, cost, and sustainability rather than raw benchmark accuracy alone.

**Broader Impact Concerns:**

The paper already includes a Broader Impact Statement and does acknowledge privacy, sustainability, and safety issues. That is a positive. My main suggestion is that the discussion should more explicitly address the fact that efficiency can enable both beneficial democratization and harmful proliferation. Lowering deployment cost can expand access, but it can also accelerate surveillance, invasive on-device sensing, and deployment in safety-critical settings without sufficient reliability guarantees. The privacy discussion should also more explicitly note that edge deployment is not automatically safe: keeping data local reduces transmission risks, but it introduces device-side extraction/tampering risks and can still enable continuous inference over sensitive multimodal streams. The current statement points in this direction, but could be more concrete.

**Claims And Evidence:**

No

**Claims Explanation:**

The paper is clearly well-read and broadly informed, and many of its descriptive claims are supported in the ordinary survey sense through representative citations, figures, and category-level discussion. The MAS structure is clearly articulated, and the paper provides concrete examples under each layer, plus complexity and representational trade-off discussions in places such as Table 1. That part is convincing.

However, I do not think the stronger claims are fully supported by the standard implied by the paper’s framing. In particular, the claims of being the first structured model-to-system taxonomy, of offering a formal foundation for EML, and of elucidating how cross-layer co-design contributes to the “Efficiency-Utility-Privacy” trade-off are not supported by a systematic comparative analysis against prior surveys or by a reproducible survey methodology. The paper states the novelty and breadth of the contribution but does not clearly explain how papers were selected, what inclusion/exclusion criteria were used, how coverage was validated, or how this survey differs rigorously from recent surveys on efficient VLMs/MLLMs/resource-efficient foundation models.

Relatedly, much of the cross-layer synthesis is insightful but still largely narrative. The survey repeatedly emphasizes “vertical synergy,” but there is no standardized evaluation template or meta-analysis showing how specific model-level choices interact with algorithmic and system-level techniques across concrete settings. Likewise, several future-facing statements, especially around intrinsic efficiency and self-regulating intelligence, are interesting perspectives but are more speculative than evidentially demonstrated. So my issue is not that the paper is inaccurate overall; it is that the most ambitious claims are only partially substantiated.

**Requested Changes:**

1. Critical - Add a much clearer survey methodology section: The paper should explicitly state how the literature was collected, what sources/time period were covered, what the inclusion and exclusion criteria were, and how the authors determined the set of “representative” papers. Right now, the survey reads knowledgeable, but not sufficiently systematic for strong novelty and comprehensiveness claims.
2. Critical - Substantiate the novelty claim relative to prior surveys: The claim that this is the first structured model-to-system taxonomy should be defended carefully. I would like to see a dedicated comparison table against prior surveys on efficient VLMs, MLLMs, and resource-efficient multimodal foundation models, showing exactly what those surveys cover and what this paper newly adds.
3. Critical - Tone down or better justify the strongest conceptual claims: Phrases such as “formal foundation,” “mitigates the Efficiency-Utility-Privacy trilemma,” and “self-regulating intelligence” currently feel stronger than the evidence supports. Either these claims should be softened and clearly framed as perspective/opinion, or the paper should provide more concrete evidence and argumentation for them.
4. Critical - Strengthen the synthesis beyond categorical organization: The paper would benefit from one or more summary tables that compare methods across shared practical axes such as training vs inference setting, modality scope, hardware assumptions, latency target, memory savings, and accuracy trade-offs. Right now many sections are broad but still somewhat enumerative.
5. Important but not strictly critical - Clarify coverage boundaries and avoid overreach. The scope section says orthogonal topics are included only when directly tied to efficiency, but in several places, the narrative drifts into broader multimodal model evolution. Tightening the boundary between efficiency-centered review and general multimodal background would improve focus.
6. Important but not strictly critical - Improve discussion of benchmarks and evaluation gaps: The applications section lists many benchmarks, but the survey could do more to discuss how efficiency should actually be measured across these settings, including latency, throughput, memory, energy, context length, robustness under compression, and deployment realism.
7. Important but not strictly critical - Address uneven granularity across sections: Some parts, especially model and algorithm sections, are very detailed, while others remain at a higher level. A more consistent depth across sections, or a short explanation of why some areas are treated more extensively, would help.
8. Minor - Clean up presentation issues and wording: There are a few phrasing issues and some terminology that would benefit from polishing. Also, some figure/table captions could be made more explicit about what is descriptive versus what is an author-synthesized conclusion.

---

> ### Author Response · Authors · 2026-04-06
> **[Part 1/2] Responses by Authors**
>
> Dear Reviewer nC85,
>
> Thank you for taking the time to provide such insightful and constructive comments. We have carefully addressed each of your suggestions in the updated manuscript, with all modifications explicitly marked in red. Our brief responses are provided below:
>
> **[Q1] Add a much clearer survey methodology section**
>
> **Response:** We fully agree that a reproducible methodology is essential for a survey of this scale.
>
> * We have added a dedicated **Section 1.3: Survey Methodology** in the Introduction. In particular, we explicitly document that our literature search covered the period from 2020 to 2026, utilizing databases including Google Scholar, arXiv, and DBLP, with a primary focus on top-tier venues (e.g., ICLR, ICML, NeurIPS, CVPR, ICCV, ECCV, ACL, EMNLP, NAACL, ACM MM, and AAAI). As reflected in our bibliography, the updated manuscript now synthesizes over  300 foundational and recent papers.
>
> * We strictly defined our inclusion criteria: papers must demonstrate quantifiable efficiency gains (e.g., FLOPs, latency, memory, or energy reduction) in a multimodal context. Works focusing solely on unimodal efficiency or general multimodal accuracy without resource optimization were excluded.
>
> * It is worth noting that A core objective of this survey is to map the structural design space of EML rather than compile an empirical numerical leaderboard. Because raw performance metrics (e.g., latency or memory reduction) are highly entangled with heterogeneous hardware backends, task formulations, and deployment constraints, direct numerical comparisons are often scientifically incomparable and misleading. Consequently, our synthesis prioritizes methodological dimensions and operational alignments.
>
> The revision is in red.
>
> **[Q2] Substantiate the novelty claim relative to prior surveys**
>
> **Response:** We appreciate the suggestions for clearer positioning. We have introduced a **Comparison Table in Appendix C (Table 5)**, comparing our work against recent excellent surveys on VLMs, MLLMs, and resource-efficient foundation models. The table explicitly highlights that while prior surveys excel in categorizing task-specific applications or isolated granular mechanisms, our work is the first to introduce a vertical, full-stack Model-Algorithm-System (MAS) taxonomy. Notably, our survey bridges the critical gap between high-level architectural designs and underlying physical execution constraints (e.g., KV cache management, hardware-aware scheduling, and edge-cloud orchestration), offering a unified blueprint for Efficient Multimodal Learning (EML) that is absent in prior literature. The revision is in red.
>
> **[Q3] Tone down the strongest conceptual claims, such as "formal foundation," "mitigates the Efficiency-Utility-Privacy trilemma", and “Self-regulating intelligence”.**
>
> **Response:** We fully agree with the reviewer that these phrases, while conceptually exciting, read too much like facts rather than forward-looking syntheses. We have carefully recalibrated our tone throughout the manuscript to ensure all claims are appropriately grounded:
>
> * **"Formal foundation":** We have softened this phrasing, **replacing it with "establishes a structured framework"** to better reflect the synthesizing nature of a survey.
>
> * **"Efficiency-Utility-Privacy trilemma":** We have revised our discussion to clarify that this is an inherent, practical deployment challenge rather than a newly coined theoretical concept. We now frame this section explicitly as a re-evaluation of existing industrial and academic tensions through the lens of our MAS framework.
>
> * **"Self-regulating intelligence":** To rigorously justify this claim, we now strictly tie it to recent empirical evidence. Specifically, we ground it in the current architectural shift toward adaptive execution graphs and elastic resource orchestration, as observed in cutting-edge models like MiniCPM-V and dynamic MoE architectures.
>
> The revision is in red.
>
> **[Q4] Strengthen the synthesis beyond categorical organization**
>
> **Response:** We completely agree that synthesis across shared axes is vital. However, a significant challenge in EML is the heterogeneity of evaluation environments (e.g., varying hardware backends, software stacks, and batching strategies). Reporting raw performance numbers (e.g., latency in ms or throughput) from disparate papers is scientifically precarious and can be highly misleading.
>
> To provide a rigorous and fair synthesis, we have significantly expanded the **Multi-dimensional Attribute Synthesis in Appendix B (Tables 3 & 4).** We categorized representative works across unified operational axes: Backbone, Parameter constraints, Efficiency Strategy (within the MAS framework), and Modality Scope. This qualitative-to-quantitative mapping provides a reliable "Design-Space Map" for practitioners, fulfilling the need for empirical synthesis without falling into the trap of biased numerical comparisons. The revision is in red.

---

> > ### Author Response · Authors · 2026-04-06
> > **[Part 2/2] Responses by Authors**
> >
> > **[Q5] Clarify coverage boundaries and avoid overreach**
> >
> > **Response:** We appreciate this constructive feedback. We have conducted a thorough review of Section 2 (Model), Section 3 (Algorithm), and Section 4 (System) to prune general multimodal background information that drifted from our core theme. We have ensured that every architectural or algorithmic discussion is now strictly contextualized by its contribution to resource optimization (e.g., computational parsimony or memory reduction). Furthermore, to explicitly establish our boundaries upfront, we have detailed our precise **scope and selection criteria** in the newly added **Subsection 1.3 (Survey Methodology)** in the Introduction. **The revision is in red.**
> >
> > **[Q6] Improve discussion of benchmarks and evaluation gaps**
> >
> > **Response:** We thank the reviewer for this insightful comment. We completely agree that evaluating EML across diverse deployment settings is a significant challenge. To address this, we have added **Subsection 8.6 (Toward Standardized Benchmarks and Evaluation)**. While we acknowledge the baseline utility of theoretical metrics like FLOPs, we explicitly argue they are insufficient for capturing memory bottlenecks and hardware overheads. To move the field forward, we advocate for comprehensive, architecture-agnostic frameworks (akin to an "MLPerf for Multimodal Efficiency"). Specifically, we propose moving toward deployment-aware metrics such as **state-update latency** for continuous models, **perception-action latency** for embodied AI, and **energy-per-sample** for edge devices. **The revision is in red.**
> >
> > **[Q7] Addressing Uneven Granularity Across Sections (System vs. Model/Algorithm)**
> >
> > **Response:** We appreciate the reviewer's observation. We wish to clarify that the relatively thinner volume of the System section is not an oversight of our survey, but rather an accurate reflection of the current EML research landscape. Until very recently, the community has overwhelmingly prioritized model architecture and algorithmic compression, leaving multimodal-specific system orchestration in its infancy.
> >
> > To address this constructively, we have:
> >
> > * **Integrated the latest advancements:** We substantially expanded Section 4 (System) with cutting-edge 2024–2026 works, diving into multimodal KV Cache management (e.g., FastCache, MPIC) and asynchronous pipelining (e.g., RServe).
> >
> > * **Framed the gap as an emerging trend:** Rather than artificially padding the section, we use this objective "unevenness" in literature to highlight a critical future direction. We now explicitly discuss in Sections 4 and 8 that bridging the gap between theoretical compression and physical system orchestration is the most urgent emerging trend for the community.
> >
> > **The revision is in red.**
> >
> > **[Q8] Clean up presentation issues and wording. There are a few phrasing issues**
> >
> > **Response:** Thanks for the detailed comments. We have gone through the whole paper and polished all the sentences.
> >
> >
> > **[Q9] Broader Impact Concerns**
> >
> > **Response:** We deeply appreciate this nuanced perspective on security and ethics. The reviewer raises a crucial point: the democratization of AI through efficiency is fundamentally a double-edged sword.
> >
> > To address this, we have substantially expanded **Subsection 8.5 (Privacy-Aware Efficiency and Security)** to explicitly acknowledge these dual-use risks and threat vectors. Specifically, we incorporated two key discussions:
> >
> > * **The Proliferation Risk:** We explicitly note that lowering the computational barrier inadvertently makes it cheaper and easier for malicious actors to deploy harmful multimodal applications (e.g., deepfakes or automated misinformation) at scale.
> >
> > * **Edge Device Vulnerabilities:** We dispel the common misconception that local/edge deployment guarantees safety. We now explicitly highlight risks inherent to physical hardware, such as device-side model extraction, side-channel attacks, and data tampering.
> >
> > By framing efficiency not just as an engineering goal but as a variable that alters the threat landscape, we believe the revised section provides a much more responsible and comprehensive view of EML's broader impacts.
> >
> > **The revision is in red.**
> >
> > Thanks again for your invaluable feedback to improve our work. We hope our revision aligns with your expectations.

---

### Review · Reviewer_ogc8 · 2026-03-23

**Summary Of Contributions:**

The paper is giving an overview of the main trends on different aspects for designing the multimodal efficient models, trying to systematize them and proposing the view of three steps (models - what to compute , algorithm - how to compute and system - where to compute) which should be considered as pieces of the entire system and optimized all together, rather than just focusing on specific component optimization, as at deployment they all will interplay in specific way, which can be optimized for efficiency and real time only when considered from the lens of the entire system. Authors cover with overview of main developments of 2-3 years in each step with details on specific main trends which the community developed. Also authors tie their framework to the recent changes and shifts in the community showing that more works and research are aligned (general focus of recent papers or example of multimodal LLMs development and recent deployments) or even precluded it (some case study in Appendix).

So in summary contributions are:
- Overview of recent trends on different aspects for building efficient multimodal systems
- Framework on model, algorithm, and system, which authors stress need to be optimized together or cross level
- Authors advocate for "paradigm shift toward self-regulating intelligence, where efficiency is an intrinsic, emergent property of the model’s fundamental design rather than a post-hoc constraint".

**Additional Comments:**

Let me know if something is unclear or some pointers are needed from speech-audio-text domain which I can point out.

**Audience:**

Yes

**Audience Explanation:**

The paper covers broad recent research on multimodal efficient models, and will be interesting for many people from various domains as a solid overview where they can start for approaching efficient multimodal models. The most important thing for me for the audience is selling the necessity of research to do full stack optimization, where any improvement in the particular place of a big system is considered within that big system and we really work towards important pieces otherwise a lot of developments and optimizations will end up w/o any deployment. Given how big models are, how much resources to train them, how many capabilities we want models to have, this focus and selling in the paper are critical.

**Broader Impact Concerns:**

All good to me, only one thing I would discuss a bit more is differential privacy as a mathematical strong guarantee for privacy.

**Claims And Evidence:**

Yes

**Claims Explanation:**

Overall I enjoyed reading and argumentation, reasoning, ideas and proposed paradigm are solid and I agree with the authors position here. However, I feel a bunch of things are missing and focus is blurred / biased towards vision-text oriented works. Please find details on things I would like to see improved in the paper before publication in the "Requested changes".

**Requested Changes:**

Conceptual questions / comments:
- First of all I would argue that the same paradigm and 3 hierarchical levels (model, algorithm, system) should be considered for unimodal models too (given that we have foundation models for specific domains and these models are big too). And deployment of multimodal models has a lot of similar issues / needs as unimodal + extra multimodality specific things. After reading it seems like I don't need to do anything for unimodal models, however most methods in every level presented are used and initially built in unimodal scope. This gives a really wrong perception, creating biases towards multimodal systems being special. In my view they expand on top of unimodal with interesting complicated and challenging new problems, but part of the problems can be solved (and solved) from unimodal developments.
- Placing federated learning (FL) under the System is really questionable to me. I would argue that it expands all three levels: specific modeling, specific algorithms and specific system optimizations.
- Most parts of the paper are with examples and references which are biased towards text-image works. There are many developments in speech, general audio, video, text models. Either scope as review from image-text side, or expand on other domains references. I am more in-depth working in speechLLMs and really there are many interesting developments that happened last year and because speech is really real time applications you need to be low latency, so there are many co-design choices needed which is a good example to advocate for your framework.
- The biggest part which is missing for me is:
  - First is introducing general concept (like you do) but with proper references on initial works as people may want to read initial formulation of e.g. KD, pruning, quantization.
  - discuss what / how it is used in unimodal works before as most all of the concepts you discuss originally were applied to unimodal case
  - then discuss and extend to multimodal works
  - finally discuss the differences between unimodal and multimodal cases: what exactly e.g. new things are done for multimodal (not just apply exactly same), why, and why this is important, expanding on the key challenges for multimodal and why in unimodal we didn't think of that problems.
  - This way I can understand why multimodal is challenging, what is special, what special remedy / design it is needed. Otherwise right now most of the concepts for me are like "yep, sure, it was done in unimodal across different modalities, nothing special, just apply as before, why then is multimodal so special? maybe it is the same?" Some minor aspects were covered in text - e.g. misalignment of modalities, different time to pre-encode and thus heterogeneous compute pipelines which bubbles. But it is not systematic and not clear after reading the paper.
- How about discussing alternative models and tokenization (not discrete!) in Sec 9.1? In the end maybe we should build not LLM based things, which may become obvious if we look at the whole system we need to make efficient from the design principles. And exactly your point on looking into the whole system rather than one component can bring people to build entirely new ones.
- For Federated learning discussion I suggest to revise section to include differential privacy, as federated learning on its own doesn't guarantee user privacy, and given that you have a lot discussions on privacy aspect (not only safety, or on-device only processing, but mix of on-device / server side) then it is a must thing.
- Many subsections on the concepts have references only to image-text models, I feel it is biased and really doesn't reflect the state of the research community. Yep, I understand that you don't cover everything which is ok, but it should be diverse, otherwise it feels we made progress only in image-text... Even System optimizations just made for image-text, but speech and video e.g. are real time low latency applications which are actually way more challenging.
- Sec 3.2 - it is unclear what is efficient encoder - data efficient or parameter efficient or inference efficient? What about references to QWEN and AudioFlamingo3 or other similar models not image-text only oriented? Also when you write "they tokenize all modalities into single discrete" it is questioned how exactly tokenization is done, as it may hide a lot of compute.
- On KD discussion - why mainly CLIP only models examples? There are soooo many things people did with KD but all works are focused on CLIP. On quantization - destabilization for vision-text models - but in fact it happens in unimodal models too.
- Pruning discussion Sec 4.2 - I would be more specific if it is structural pruning or not for the last sentence, as in general pruning may be inefficient for hardware if not designed with it in mind.
- For quantization discussion Sec 4.3 "accelerates inference" I would be specific how and why it speeds up inference, because it is tied exactly to hardware. In general I see that many concepts are actually in between levels (like they both algorithm and system or model and algorithm) - so maybe make this discussion clear that if we optimize - we get some speed up maybe, in reality not, or only if specific hardware or implementation is used? I feel this clear explanation of how a specific concept on one level can be fast or slow for the next level is missing as a pointer for people where they should look.
- Still Sec 8 for me can be applied to unimodal models too.... I really struggle to see specifics created by multimodal systems from reading the current draft.
- Page 9 "Efficiency as a Structural Prerequisite Not a Trade-off:" - most works still do not design in this way really...

General questions / comments:
- I did not get the difference between modality specific encoders and unified encoders. In most works you provide it is still modality specific shallow encoder, thus it is not really unified. If we make a definition that transformer layers are not presented in encoders - this is a wild definition for me. I would argue it is more about how many compute we spend on modality specific thing vs using compute on joint processing - but this is not discussed that way and even not provided how unified encoders actually more efficient - e.g. no model sizes, no FLOPs / runtime, no info on how deep models are - as likely you need deeper stack on processing and thus it is not efficient really. There are works which treat every pixel, audio point, text as a token - this will be unified to me.
- Page 6 "As decoder-only LLMs like LLama..." why LLama, not even GPT which were the first, or other recent models?
- Page 6 - Audio section is fully covered only in general audio space except SSL for speech. I think it is not reflecting proper Audio-Speech domain, where speech is a crucial modality for conversation and it is not covered adequately for recent speech-text, speech-text-video models at all. Please revise properly.
- Section 3 - why no discussion on video?

Minor on the presentation:
- Fig 1. Compression is used in Models, but then mentioned in the caption for Algorithms - this is confusing. Overall I found naming Algorithm to be confusing - a bunch of things can be considered as modeling and vice versa... Maybe more precise naming / definition can be provided earlier (only later in the end of the paper I kind of got your meaning / separation). E.g. I could say MOE is also an algorithm... Also Applications are not discussed in the caption, though they are the fourth part...
- I feel there are many repetitions of the main messages in Introduction and Scope and Taxonomy and later in every section. I suggest merging Sec 1 and 2.
- "offering superior scaling laws" on page 5 - I found it weird in the context, as O(N) doesn't say anything about scaling law. I think the usage of "scaling law" is not appropriate here.
- in many references like "in work(Person et al)" there is missing space
- Appendix case study: for results in A1, A2, A3 (specifically latency, energy, storage) are they from the prior work or you did this evaluation / optimization? it is unclear from the text.
- Please add references to the main methods / concepts of prior works, like KD, pruning, quantization, kernels optimization, kv cache, secure aggregation, fedavr, federated learning, etc. All the things are w/o any references / pointers. This may be ok if people are very familiar with all standard methods by now, but this is a survey paper - so good to be very specific for people from different domains if they wanna expand cross domain / whole system knowledge and build such systems in future (for which you advocate).

---

> ### Author Response · Authors · 2026-04-06
> **[Part 1/2] Responses by Authors**
>
> Dear Reviewer ogc8,
>
> We sincerely thank you for your detailed and constructive feedback. We have carefully revised the manuscript based on your suggestions. Please find our responses below (all revisions are highlighted in red in the revised manuscript):
>
> **[Q1] Unimodal vs. Multimodal Distinction**
>
> **Response:** We appreciate this insightful comment. We agree that the technologies are universal if we dive into the basic principles. Considering the unique challenges should be introduced early in this survey so that all the readers, especially the readers from other domains, can have a big picture of the Model-Algorithm-System structure, we highlight the unique challenges by contrasting Unimodal and Multimodal in **Subsection 1.1** in the revised version, which is also highlighted in Red. In particular, we highlight unique multimodal challenges such as cross-modal alignment overhead, modality-specific synchronization, and inference pipeline "bubbles" caused by heterogeneous input processing (e.g., the time difference among encoding a high-resolution video, a short text prompt, and a long audio sequence). Besides, for the remaining sections, we also mention the unique challenges for efficient multimodal if needed, such as Section 2, Section 3, and Section 4.
>
> **[Q2] Categorization of Federated Learning (FL)**
>
> **Response:** We agree that FL is indeed a cross-layer paradigm. We will move the FL part to the Application Section to reflect that FL involves Model partitioning (e.g., Split Learning), Algorithmic optimization (e.g., FedAvg/FedProx), and System-level communication efficiency. FL is a good application case to demonstrate the contribution of cross-layer optimization, which considers the Model-Algorithm-System co-design to make the whole pipeline more efficient and privacy-reserved. The revision is in red.
>
> **[Q3] Domain Bias (Vision-Text focused. Please add more references like Audio-Speech-Video modalities.)**
>
> **Response:** Thanks for the useful suggestions. Audio and Speech, of course, are the main modalities in our survey. We significantly expand our coverage of SpeechLLMs (e.g., Qwen-Audio, SALMONN) and Video-centric models. Specifically, we discuss Speech-focused efficient multimodal to highlight the importance and necessity and efficient multimodal learning again, as speech interaction requires stricter real-time constraints than static image analysis.
>
> **[Q4] Missing Foundational References for KD, Pruning, and Quantization.**
>
> **Response:** We rectify this in the revision by including foundational citations, such as Hinton et al. (2015) for Knowledge Distillation, Han et al. (2015) for Deep Compression, and Lloyd/Max for quantization theory, ensuring the survey serves as a complete academic roadmap even if the readers are not familiar with this domain. The highlight is in red.
>
> **[Q5] Definition of Unified Encoders.**
>
> **Response:** Thanks for the detailed suggestion. In current modality-specific encoders, different modalities require different, specifically-designed encoders, such as BERT for Text, HuBert for Audio, Facet for Image, and so on. Each encoder is designed and trained separately, which generally requires 3x parameters for the encoders. Therefore, we define "Unified Encoders" that do not need to design specific encoders for each modality, but just need to vectorize all the modalities so that they can be processed by a unified backbone. As shown in Figure 3, this classification is clear and intuitive. We also provide a clearer definition for the unified encoders, which is highlighted in red.
>
> **[Q6] Alternative Architectures & Non-discrete Tokenization**
>
> **Response:** Thanks for the insightful suggestions. We have added a discussion on continuous representation learning and alternative architectures (e.g., state-space models like Mamba) that might bypass the quadratic bottleneck of Transformer-based discrete tokenization in multimodal contexts, which is discussed in Section 9.1 as highlighted in red.
>
> **[Q7] Differential Privacy (DP) should be discussed with FL.**
>
> **Response:** We moved the Federated Learning part to the Application Section, which will comprehensively involve methods in modal, algorithm, and systems levels. We also incorporate discussion on Differential Privacy (DP) in the Federated Learning part, highlighting it as a mathematical guarantee that complements FL in protecting sensitive user data across modalities. All the discussion is highlighted in red.
>
> **[Q8] Detailed Mechanisms of Pruning**
>
> **Response:** In the Pruning subsection, we edited so that the section discusses unstructured and structural pruning, and clarify that Structural Pruning is prioritized for hardware compatibility for efficiency. The revision is in red.

---

> > ### Author Response · Authors · 2026-04-06
> > **[Part 2/2] Responses by Authors**
> >
> > **[Q9] Detailed Mechanisms of Quantization**
> >
> > **Response:** For quantization, we explain the hardware compatibility issue associated with quantization, such as reduced memory bandwidth pressure and the utilization of specialized integer arithmetic units (e.g., INT8 Tensor Cores). All the detailed discussions are highlighted in red.
> >
> > **[Q10] Scaling Law Terminology, Using "superior scaling laws" for $O(N)$ complexity is inappropriate.**
> >
> > **Response:** We apologize for the terminological inaccuracy. We replace this with "linear computational scalability" regarding sequence length to accurately reflect the complexity benefits. The revision is in red.
> >
> > **[Q11] Figure 1 naming is inconsistent with the text; Sec 1 and 2 are repetitive.**
> >
> > **Response:** First, we update Figure 1, changing the “adapter compression” to “modular adaptation”, and ensure consistent terminology throughout the manuscript. Second, we will merge Sections 1 and 2 for better flow. Besides, we will add a new subsection 1.1 to highlight the unique challenges of efficient multimodal learning compared to unimodal learning.  All the revisions are in red.
> >
> > Thanks again for your invaluable feedback to improve our work. We hope our revision aligns with your expectations.

---

> > > ### Comment · Reviewer_ogc8 · 2026-04-18
> > > **Comments on the revision**
> > >
> > > Dear Authors,
> > >
> > > Finally I did a pass over the paper. Looks really good to me and reads smoothly now! Also thanks a lot for including speech / video more extensively throughout the paper. And I like connection to unimodal and showcasing the challenges specific to multimodal models at every layer.
> > >
> > > I have only minor comments which will be good to include in camera ready from my perspective:
> > > - Figure 1: typo in "Strctural sparsity"
> > > - Page 7, audio subsection - I still will include here references not only on SSL encoders, but also discrete / continuous tokenizers, which now are used as speech encoders: Whisper encoder (is extensively used now in speechLLMs to plugin speech into text-speech and video-text-speech models), SpeechTokenizer, Mimi (as famous from Moshi).
> > > - page 9: add citations to EnCodec and Video, 3D tubelet.
> > > - Table 1: can we make general 2 domains instead of visual + text tokens? or maybe add "e.g.". I see that except "Token Reduction (e.g., ToMe)" everything can be applied to arbitrary 2 domains.
> > > - page 15: "alignment between vision and language" just say "alignment between modalities"? as you are talking about general methods, so we might not need to tight to specific modalities?
> > > - page 16: repeated sentence "These methods provide a lightweight baseline for replicating reasoning behavior through output imitation. These methods provide a lightweight baseline for transferring teacher decisions through output imitation,"
> > > - page 18: "While textual tokens are semantically dense and sequential, visual tokens" - can we change it to "While textual tokens are semantically dense and sequential, e.g. visual / audio tokens" - maybe add more other modalities with similar property?
> > > - page 21: maybe change "where the large language model (LLM) often sits idle waiting for the visual encoder to process high-resolution images." to "where the large language model (LLM) often sits idle waiting for other modalities encoders to process high-resolution streams." to make more general?
> > > - page 22: "Solutions like RServe" -> add citation?
> > > - thanks for moving federated learning to applications. "This functionality is essential for privacy-critical applications where data sovereignty is mandated by legal and ethical boundaries." As I said before, **federated learning doesn't bring privacy, so I still insist on adding here just one sentence "in combination with differential privacy [citation]".**
> > > - page 30: "such as secure aggregation" -> add citation, "compressed encrypted inference" -> add citation.
> > >
> > > And one general thing - maybe it is obvious, but I thought about it while reading:
> > > - what multimodal models do you have in mind: input is multimodal, output is multimodal, both? also is it necessary that multimodal data are paired? or it can be fully disjoined sets (set of images, set of text)? Maybe good to add in the intro clarification just in case. Likely you cover all cases here?
> > >   - Asking this because e.g. for understanding tasks and generation tasks you have different optimizations and efficiency prerequisites. Also encoders e.g. will be different, and maybe you then talk also about detokenization (e.g. for video, speech, maybe images). But maybe it is too specific and deep which is not needed :).
> > >
> > > Thanks again for reworking the paper!
> > >
> > > Reviewer.

---

> > > > ### Author Response · Authors · 2026-04-18
> > > >
> > > > **Dear Reviewer ogc8,**
> > > >
> > > > We sincerely thank you for the encouraging positive feedback. We are thrilled that you find the paper reads smoothly and appreciate the expanded coverage of speech/video and the connection to unimodal challenges.
> > > >
> > > > All final minor comments are incredibly precise and have further polished the rigor and generalizability of our manuscript. We have carefully addressed each of your suggestions in the revised version as detailed below: (All revisions are in red in the updated version.)
> > > >
> > > >
> > > > **[Q1] Figure 1: typo in "Strctural sparsity"**
> > > >
> > > > **Response:** Thanks. We have corrected the typo to "Structural sparsity" in Figure 1.
> > > >
> > > > **[Q2] Page 7, audio subsection. References, such as Whisper encoder, SpeechTokenizer, Mimi (as famous from Moshi).**
> > > >
> > > > **Response:** Thanks. Following the reviewer's suggestion, we have expanded the audio subsection to explicitly cover these advancements. We added discussions and citations for Whisper (Radford et al., 2023), SpeechTokenizer (Zhang et al., 2023) and Mimi (Défossez et al., 2024).
> > > >
> > > > **[Q3] page 9: add citations to EnCodec and Video, 3D tubelet.**
> > > >
> > > > **Response:** Thanks. We cited EnCodec (Défossez et al., 2022) for neural audio codecs, ViViT (Arnab et al., 2021) for 3D tubelet embeddings to support the discussion on interleaving discrete video indices with text tokens.
> > > >
> > > > **[Q4] Table 1: can we make general 2 domains instead of visual + text tokens? or maybe add "e.g.". I see that except "Token Reduction (e.g., ToMe)" everything can be applied to arbitrary 2 domains.**
> > > >
> > > > **Response:** This is an excellent point. We have updated the Table 1 caption to generalize the formulation beyond just visual and text tokens. The caption now defines $N$ and $M$ generally: "$N$ and $M$ denote the lengths of the token sequences for two distinct modalities, such as vision and language, respectively." This accurately reflects that these complexity bounds apply to arbitrary cross-modal interactions.
> > > >
> > > > **[Q5] page 15: "alignment between vision and language" just say "alignment between modalities"?**
> > > >
> > > > **Response:** Thanks for the suggestion. We changed "alignment between vision and language" to "alignment between modalities" to ensure the discussion remains general across all heterogeneous spaces.
> > > >
> > > > **[Q6] page 16: repeated sentence "These methods provide a lightweight baseline for replicating reasoning behavior through output imitation."**
> > > >
> > > > **Response:** We apologize for that. We have removed the duplicated sentence and ensured the paragraph flows smoothly.
> > > >
> > > > **[Q7] page 18: "While textual tokens are semantically dense and sequential, visual tokens" - can we change it to "While textual tokens are semantically dense and sequential, e.g. visual / audio tokens" - maybe add more other modalities with similar property?**
> > > >
> > > > **Response:** Thanks for the nice suggestion. The sentence has been revised to: "While textual tokens are semantically dense and sequential, tokens from other modalities---such as visual and audio streams---often exhibit high spatial and temporal correlation."
> > > >
> > > > **[Q8] page 21: maybe change "where the large language model (LLM) often sits idle waiting for the visual encoder to process high-resolution images." to "where the large language model (LLM) often sits idle waiting for other modalities encoders to process high-resolution streams." to make more general?**
> > > >
> > > > **Response:** Thanks. We have revised the sentence accordingly: "...where the large language model (LLM) often sits idle waiting for other modalities encoders to process high-resolution streams," successfully generalizing the cases beyond just vision.
> > > >
> > > > **[Q9] page 22: "Solutions like RServe" -> add citation?**
> > > >
> > > > **Response:** We have added the citation for RServe (Guo et al., 2025a).
> > > >
> > > > **[Q10] Federated learning doesn't bring privacy, adding here just one sentence "in combination with differential privacy [citation]".**
> > > >
> > > > **Response:** We appreciate the precise suggestion. We have explicitly stated this necessity: "While FL inherently addresses data sovereignty mandates by localizing raw data, achieving true protection against inference attacks requires it to be used in combination with differential privacy (Geyer et al., 2017; Wei et al., 2020)."
> > > >
> > > > **[Q11] page 30: "such as secure aggregation" -> add citation, "compressed encrypted inference" -> add citation.**
> > > >
> > > > **Response:** We have added the foundational citations as requested.
> > > >
> > > > **[Q12] what multimodal models? input is multimodal, output is multimodal, both? also is it necessary that multimodal data are paired? or it can be fully disjoined sets (set of images, set of text)?**
> > > >
> > > > **Response:** This is an incredibly insightful comment. To clarify this, we have added a dedicated clarification within "The Multimodal Efficiency Challenge" subsection in the Introduction 1.1.
> > > >
> > > > In summary, we sincerely thank the reviewer for their valuable time and constructive feedback, which have significantly strengthened our manuscript.
> > > >
> > > > Authors

---

> > > > > ### Comment · Reviewer_ogc8 · 2026-04-18
> > > > >
> > > > > Thank you!

---

> > > > > > ### Author Response · Authors · 2026-04-26
> > > > > >
> > > > > > Dear Reviewer ogc8,
> > > > > >
> > > > > > Thank you so much for your time and effort on our work. Your insightful and constructive suggestions were incredibly valuable, which have significantly helped us improve the comprehensiveness and overall quality of this survey, making it a much better reference for the community.
> > > > > >
> > > > > > Best wishes to you.
> > > > > >
> > > > > > Thanks,
> > > > > >
> > > > > > Authors

---

### Decision · Action_Editor_A2YM · 2026-04-28

**Recommendation:** Accept as is

**Audience:**

Yes

**Audience Explanation:**

The paper covers broad recent research on multimodal efficient models, and will be interesting for many people from various domains.

**Claims And Evidence:**

Yes

**Claims Explanation:**

All reviewers agreed that this survey is a high-quality summarization of this direction, and recommended acceptance. The revision is good, which addresses many previous concerns.